# Dysregulation of Wnt/β-catenin signaling contributes to intestinal inflammation through regulation of group 3 innate lymphoid cells

Jiacheng Hao[1,2,3,4,5], Chang Liu[2], Zhijie Gu[1,2,3,4,5], Xuanming Yang ![ORCID][6,7,8], Xun Lan ![ORCID][2,5] & Xiaohuan Guo ![ORCID][1,2,4] ✉

RORγt+ group 3 innate lymphoid cells (ILC3s) are essential for intestinal homeostasis. Dysregulation of ILC3s has been found in the gut of patients with inflammatory bowel disease and colorectal cancer, yet the specific mechanisms still require more investigation. Here we observe increased β-catenin in intestinal ILC3s from inflammatory bowel disease and colon cancer patients compared with healthy donors. In contrast to promoting RORγt expression in T cells, activation of Wnt/β-catenin signaling in ILC3s suppresses RORγt expression, inhibits its proliferation and function, and leads to a deficiency of ILC3s and subsequent intestinal inflammation in mice. Activated β-catenin and its interacting transcription factor, TCF-1, cannot directly suppress RORγt expression, but rather alters global chromatin accessibility and inhibits JunB expression, which is essential for RORγt expression in ILC3s. Together, our findings suggest that dysregulated Wnt/β-catenin signaling impairs intestinal ILC3s through TCF-1/JunB/RORγt regulation, further disrupting intestinal homeostasis, and promoting inflammation and cancer.

Gut homeostasis is crucial for maintaining the host's health and preventing diseases like inflammatory bowel disease (IBD) and colorectal cancer (CRC). Various signaling pathways, including the Wnt/β-catenin pathway, exert significant influence on intestinal homeostasis. Wnt/β-catenin pathway-related proteins are abundant in the gut microenvironment, and they play a crucial role in intestinal epithelial renewal[1]. Mutations or dysregulation in genes related to the Wnt/β-catenin pathway, such as *adenomatous polyposis coli* (*APC*), causes stabilization and hyperactivation of β-catenin, which initiates most

human colorectal cancer[2,3]. Growing evidence indicates the strong association of the Wnt/β-catenin pathway with IBD and colitis-associated colorectal cancer[4]. In addition to driving carcinogenesis in gut epithelial cells, dysregulated Wnt/β-catenin signaling also promotes colitis and colon cancer by imprinting proinflammatory properties in T cells[5]. The genetic activation of β-catenin in mice epigenetically reprograms T cells, induces expression of Th17 signature genes, including RORγt, and facilitates Th17 and RORγt+ Treg-mediated colonic inflammation and cancer[5-7]. However, whether

[1]Institute for Immunology, Tsinghua University, 100084 Beijing, China. [2]Department of Basic Medical Sciences, School of Medicine, Tsinghua University, 100084 Beijing, China. [3]School of Life Sciences, Tsinghua University, 100084 Beijing, China. [4]Beijing Key Lab for Immunological Research on Chronic Diseases, Tsinghua University, 100084 Beijing, China. [5]Tsinghua-Peking Center for Life Sciences, Tsinghua University, Beijing, China. [6]Sheng Yushou Center of Cell Biology and Immunology, School of Life Sciences and Biotechnology, Shanghai Jiao Tong University, 200240 Shanghai, China. [7]Joint International Research Laboratory of Metabolic and Developmental Sciences, Shanghai Jiao Tong University, 200240 Shanghai, China. [8]Key Laboratory of Systems Biomedicine (Ministry of Education), Shanghai Center for Systems Biomedicine, Shanghai Jiao Tong University, 200240 Shanghai, China. ✉e-mail: guoxiaohuan@tsinghua.edu.cn

Wnt/β-catenin signaling in other immune cells within the intestinal inflammatory or tumor microenvironment is also dysregulated and contributes to disease progression is not clear.

Innate lymphoid cells (ILCs), including ILC1s, ILC2s, and ILC3s, are lymphocytes that share similar features with their CD4[+] T helper cell counterparts[8] and play important roles in various tissues[9]. RORγt-expressing group 3 innate lymphoid cells (ILC3s) are primarily found within the gut mucosa and in gut-associated lymphoid tissues, and can further be divided into three subsets, including CCR6[-]NKp46[+] ILC3s, CCR6[-]NKp46[-] ILC3s, and CCR6[+] lymphoid tissue inducer (LTi) cells that help develop secondary and tertiary lymphoid tissues[10]. Through its action on intestinal epithelial cells, myeloid cells, and T cells, ILC3s modulate the intestinal microenvironment by producing effector molecules, such as IL-22, GM-CSF, IL-17A, IFN-γ, MHC-II, and lymphotoxin[11]. These interactions between ILC3s and other gut cells, as well as commensal bacteria, are not only crucial for the defense against various infections but also for maintaining intestinal homeostasis[12,13]. Studies have shown that ILC3s are important in the development of IBD and colorectal cancer[14–19]. A reduced number of ILC3s and dysregulated ILC3 functions have been found in the guts of patients with IBD and colon cancer[18]. However, it remains unclear how the inflammatory or tumorigenic intestinal microenvironment influences ILC3s.

Here, we show that Wnt/β-catenin signaling in ILC3s is also dysregulated in IBD and CRC patients. In line with the reduced ILC3s in human patients, dysregulated Wnt/β-catenin signaling also results in ILC3 deficiency and leads to increased intestinal inflammation in mice. In contrast to promoting RORγt expression in T cells, activated Wnt/β-catenin signaling in ILC3s inhibits RORγt expression and impairs ILC3s' function through epigenetic reprogramming and regulation of TCF-1–JunB axis, suggesting different regulation mechanisms for Wnt/β-catenin signaling in innate and adaptive lymphocytes.

## Results

### Wnt/β-catenin signaling is dysregulated in ILC3s from IBD and CRC patients

Studies have shown that ILC3s play an important role in IBD and CRC patients[16–19]. In order to examine the potential effects of the inflammatory or tumorigenic intestinal microenvironment on ILC3s, single-cell RNA sequencing (scRNA-seq) data collected from colon tissues of IBD and CRC patients were further re-analyzed[16,17]. Similar to previous reports that colonic Th17 cells and Treg cells exhibit high levels of β-catenin protein in IBD patients[5,6], the mRNA levels of Wnt/β-catenin pathway-associated genes were also increased in IBD and CRC patients (Fig. 1a). Interestingly, ILCs, particularly ILC3s, in the inflammatory or tumor microenvironment of patients with IBD or CRC expressed significantly higher levels of *CTNNB1*, and Wnt/β-catenin downstream genes *TCF7*, *VEGFA* and *MYC*, as compared with control tissues (Fig. 1a). The data suggest that Wnt/β-catenin signaling may be also dysregulated in ILC3s from inflamed colonic tissues and tumors, which might influence the homeostasis of ILC3s.

### The activation of Wnt/β-catenin signaling differentially affects ILC3s and T cells in vitro

In order to examine the effect of dysregulated Wnt/β-catenin signaling in ILC3s, mouse intestinal ILC3s were isolated and treated with CHIR-99021, an activator of Wnt/β-catenin signaling by inhibiting GSK-3β. As a control, mouse Th17 and iTreg cells were also induced in vitro and stimulated with the Wnt/β-catenin activator. In spite of the fact that the Wnt/β-catenin activator treatment increased β-catenin protein levels in both ILC3s and T cells, different effects were observed on RORγt expression in ILC3s and T cells. In contrast to promoting RORγt expression in Th17 and Treg cells, activating Wnt/β-catenin signaling in ILC3s inhibited RORγt expression (Fig. 1b–k and Supplementary Fig. 1a–e), indicating different regulatory mechanisms of Wnt/β-catenin signaling between RORγt[+] ILCs and T cells. Moreover, the

dysregulated Wnt/β-catenin signaling in ILC3s also resulted in decreased cell viability (Fig. 1l–n) and impaired IL-17A and IL-22 production (Fig. 1o–q). In order to exclude the possibility of Wnt activator toxicity, ILC2s were also isolated and treated with CHIR-99021 at the same dose as ILC3s. Despite the Wnt activator significantly increasing β-catenin protein levels, GATA-3 expression and cell viability in ILC2s were not affected (Supplementary Fig. 1f–m), suggesting that increased activation of Wnt/β-catenin has a limited effect on ILC2s. Furthermore, intestinal ILC3s were isolated and cultured with IL-2 and IL-7 in vitro, then were stimulated either with Wnt ligands (Wnt), PMA/Ionomycin (P+I), or a combination of Wnt ligands and PMA/Ionomycin (Wnt+P+I). Flow cytometry analysis showed that Wnt ligands significantly inhibited RORγt expression in PMA/Ionomycin-activated ILC3s but not control ILC3s (Fig. 1r–u), suggesting that Wnt signaling may have an effect on ILC3s during inflammation. Due to the reduced ILC3 number and impaired function of ILC3s found in patients with IBD and CRC, our data suggest that the inflammatory or tumorigenic microenvironment in these patients may affect ILC3s by dysregulating Wnt/β-catenin signaling.

### Genetically engineered activation of β-catenin leads to ILC3s deficiency in vivo

To further determine the effect of dysregulated Wnt/β-catenin signaling on ILC3s, a mouse model with genetically engineered activation of β-catenin was established through crossing *Ctnnb1*[ex3fl/fl] mice with *Rorc*[cre] transgenic mice (*Rorc*[cre]*Ctnnb1*[ex3fl/wt]). In *Rorc*[cre]*Ctnnb1*[ex3fl/wt] mice, the Cre-mediated excision of *Ctnnb1* exon 3 removes phosphorylation sites for protein degradation, resulting in constitutively active β-catenin signaling[7] in RORγt-expressing ILC3s and T cells. ILC3 numbers and functions were examined in gut and mesenteric lymph nodes (mLN) from KO (*Rorc*[cre]*Ctnnb1*[ex3fl/wt]) mice and control (*Ctnnb1*[ex3fl/wt]) mice. All ILC3s, including CCR6[+] LTi cells and NCR[+] ILC3s, were dramatically reduced in the colon, small intestine, and mLN from KO mice (Fig. 2a–d and Supplementary Fig. 2a–g). Consistently, the deficiency of Peyer's patches (PPs) was also observed in KO mice (Supplementary Fig. 2h, i), since LTi cells are essential for the development of secondary lymphoid organs. The production of cytokines was also impaired in KO ILC3s, particularly IL-22 (Fig. 2e, f and Supplementary Fig. 2j, k). These data indicate that activation of Wnt/β-catenin signaling results in the severe deficiency of ILC3s in vivo.

As RORγt is transiently expressed in double-positive (DP) thymocytes and essential for DP thymocytes' survival[20], the effects of activated β-catenin signaling on thymocytes and T cells were also examined. Genetic activation of β-catenin signaling in KO mice led to a significantly increased proportion of RORγt-expressing Treg cells, and increased, although not significant, induction of Th17 cells in the gut (Fig. 2g, h). Interestingly, the proportion of RORγt[+]FoxP3[+] cells in total FoxP3[+] Treg cells was decreased in *Rorc*[cre]*Ctnnb1*[ex3fl/wt] mice, in contrast with the in vitro results (Supplementary Fig. 1a–e). Since the development of RORγt[+]FoxP3[+] T cells in vivo is also affected by other cells, such as T cells and ILC3s, our data suggest that both intrinsic and extrinsic β-catenin-related mechanisms contribute to the development of RORγt[+] Treg cells in vivo. Additionally, enhanced RORγt expression by DP thymocytes was also observed (Supplementary Fig. 2l, m), which is consistent with previous studies[21] and demonstrates that Wnt/β-catenin signaling could indeed promote RORγt expression in T cells.

To determine whether ILC3s' deficiency in KO mice is cell-intrinsic, bone marrow cells from CD45.1[+] C57BL/6 and CD45.1[+]CD45.2[+] *Ctnnb1*[ex3fl/wt] control or CD45.1[+]CD45.2[+] *Rorc*[cre]*Ctnnb1*[ex3fl/wt] KO mice were co-transferred into irradiated *Rag2*[−/−]*Il2rg*[−/−] recipients (Fig. 2i), and ILC3s in the gut were examined 6 weeks later. Compared with ILC3s from CD45.1[+] C57BL/6 or CD45.1[+]CD45.2[+] *Ctnnb1*[ex3fl/wt] mice, ILC3s derived from CD45.1[+]CD45.2[+] *Rorc*[cre]*Ctnnb1*[ex3fl/wt] mice still exhibited a significant reduction in both small and large intestine (Fig. 2j, k). Taken together, these data suggest that activated Wnt/β-

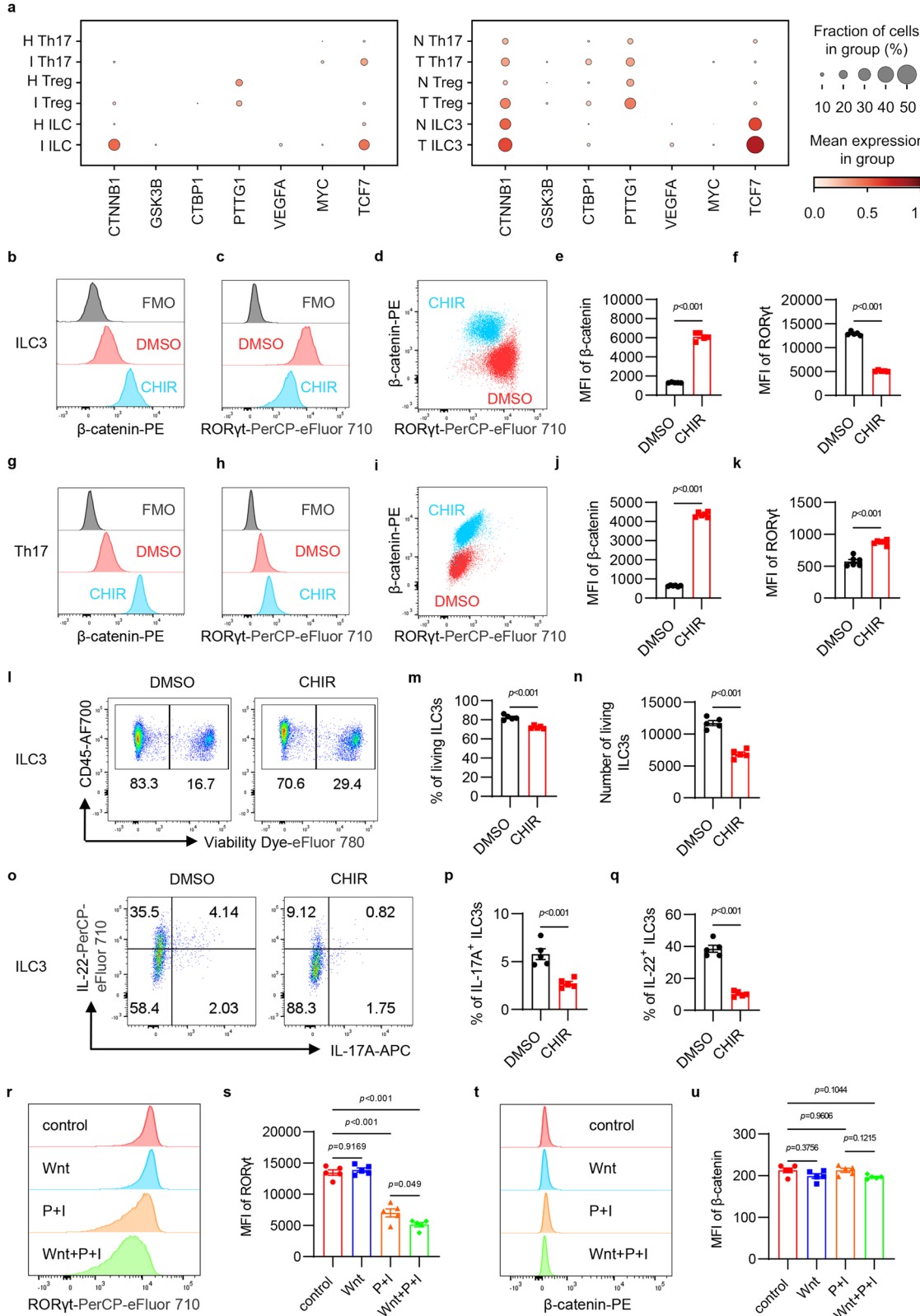

catenin signaling could cell-intrinsically suppress RORγt⁺ ILC3s, but not RORγt-expressing T cells.

## Activated Wnt/β-catenin signaling in ILC3 increases susceptibility to colitis

Next, in order to find out if activated Wnt/β-catenin signaling in ILC3s contributes to gut inflammation in IBD or CRC diseases,

*Rorc*^cre*Ctnnb1*^ex3fl/wt and control *Ctnnb1*^ex3fl/wt mice were firstly challenged with DSS-induced colitis. Upon oral administration of DSS, *Rorc*^cre*Ctnnb1*^ex3fl/wt mice rapidly lost body weight and died around day 8, compared to much less body weight loss and no deaths among their littermate control mice (Fig. 3a, b). More severe diarrhea, colitis, and colon pathology were also observed in *Rorc*^cre*Ctnnb1*^ex3fl/wt mice compared to control mice (Fig. 3c–f). Immunofluorescence staining further

**Fig. 1 | The activation of Wnt/β-catenin signaling differentially affects ILC3s and T cells in vitro. a** Relative expression level of indicated genes in T cells, ILCs from healthy (H) and inflamed (I) tissues of IBD patients, T cells, ILC3s from normal tissues (N), and colorectal tumors (T) of CRC patients. The size of the dots represents the fraction of cells in the group, and the color of the dots represents the mean expression of genes in the group. **b–f** ILC3s (gated in 7AAD⁻lineage⁻CD90^high CD45^low) were treated with DMSO and CHIR-99021 (CHIR) for 24 hours in vitro. Flow cytometry analysis of β-catenin (**b**) and RORγt (**c**) are shown (**d**). FMO represents fluorescence minus one control. Mean fluorescence intensity (MFI) of β-catenin (**e**) and RORγt (**f**) are shown. **g–k** Naïve T cells (gated in 7AAD⁻CD4⁺CD25⁻CD62L^high CD44^low) from wild type mice were sorted and Th17 cells were induced in vitro. Induced Th17 cells were treated with DMSO and CHIR for 96 hours. Flow cytometry analysis of β-catenin (**g**) and RORγt (**h**) are shown (**i**). FMO represents fluorescence minus one control. MFI of β-catenin (**j**) and RORγt (**k**) are shown. **l–q** ILC3s were treated with DMSO and CHIR for 24 hours in vitro. Flow cytometry analysis of cell viability (**l**) and cytokine production (**o**) are shown. The percentage and number of living ILC3s (**m, n**), the percentage of IL-17⁺ and IL-22⁺ ILC3s (**p, q**) are shown. Cytokine production was detected under PMA/Ionomycin treatment. **r–u** ILC3s were treated either with Wnt ligands (Wnt), PMA/Ionomycin (P+I) or a combination of Wnt ligands with PMA/Ionomycin (Wnt+P+I). Flow cytometry analysis of RORγt (**r**) and β-catenin (**t**) are shown. MFI of RORγt (**s**) and β-catenin (**u**) are shown. Each dot represents one individual replicate (*n* = 5 per group in **e, f, m, n, p, q, s, u**, *n* = 6 per group in **j, k**). Error bars represent the SEM. Statistical significance was tested by unpaired two-sided Student's *t*-test (**e, f, j, k, m, n, p, q**) and two-sided one-way ANOVA with Tukey's test adjusted for multiple comparisons (**s, u**). Data are representative of three independent experiments.

revealed that discontinuous epithelial EpCAM staining, reduced Ki-67⁺ epithelial cells and increased CD45⁺ immune cell infiltration in the colons from *Rorc*^cre*Ctnnb1*^ex3fl/wt mice (Supplementary Fig. 3a, b), indicating that over-activation of β-catenin signaling in ILC3s resulted in impaired intestinal epithelial repair and increased inflammatory responses in a DSS-induced colitis model. Consistently, *Rorc*^cre*Ctnnb1*^ex3fl/wt mice still showed reduced ILC3s numbers in the colon (Fig. 3g, h) and impaired IL-17A and IL-22 production by ILCs post DSS challenge (Fig. 3i, j). The expression of ILC3-related genes *Reg3b*, *Reg3g*, *Il17a*, *Il22* and epithelial regeneration-related marker[22] genes *Epcam*, *Tacstd2*, *Ly6a*, *Ly6g*, *Anxa1*, *Anxa8* in colon tissue was also impaired in *Rorc*^cre*Ctnnb1*^ex3fl/wt mice post DSS treatment (Supplementary Fig. 3c, d), which indicated the disorder of gut immunity and the impairment in tissue repair.

To further verify the pathogenic effect of activated Wnt/β-catenin signaling in ILC3s for colitis, wild-type ILC3s were transferred into *Rorc*^cre*Ctnnb1*^ex3fl/wt mice. The ILC3s replenishment rescued the body weight loss and death of *Rorc*^cre*Ctnnb1*^ex3fl/wt mice (Fig. 3k, l), and also significantly alleviated colon pathology and colitis (Fig. 3m, n). Additionally, *Rag1*^−/−*Rorc*^cre*Ctnnb1*^ex3fl/wt and control *Rag1*^−/−*Ctnnb1*^ex3fl/wt mice were challenged with DSS-induced colitis in order to exclude the influence of T cells. Similarly, *Rag1*^−/−*Rorc*^cre*Ctnnb1*^ex3fl/wt mice rapidly lost body weight and died around day 7, compared to control mice which only lost much less weight loss (Fig. 3o, p). More severe diarrhea, colitis, and colon pathology were observed in *Rag1*^−/−*Rorc*^cre*Ctnnb1*^ex3fl/wt mice than in control mice (Fig. 3q, r). Further, the number of ILC3s and production of IL-22 and IL-17A were all decreased in *Rag1*^−/−*Rorc*^cre*Ctnnb1*^ex3fl/wt mice (Supplementary Fig. 3e–h). Together, these data demonstrate that dysregulated Wnt/β-catenin signaling causes ILC3s deficiency and thereby increases susceptibility to DSS-induced colitis.

To further determine the effect of activated Wnt/β-catenin signaling in ILC3s on gut inflammation, the *Citrobacter rodentium* infectious colitis model was also established with *Rorc*^cre*Ctnnb1*^ex3fl/wt and control mice. Similar to the DSS-induced colitis model, rapid body weight loss and deaths were observed in *Rorc*^cre*Ctnnb1*^ex3fl/wt mice after oral *C. rodentium* infection, in contrast to no body weight loss or deaths of their littermate control mice (Supplementary Fig. 4a, b). *Rorc*^cre*Ctnnb1*^ex3fl/wt mice exhibited much higher fecal pathogen loads (Supplementary Fig. 4c), systemic dissemination of *C. rodentium* as shown by elevated blood pathogen titers (Supplementary Fig. 4d), and more severe colitis and colon pathology than *Ctnnb1*^ex3fl/wt mice (Supplementary Fig. 4e–h). Furthermore, flow cytometry showed reduced ILC3s numbers and impaired IL-22 production by ILC3s in *Rorc*^cre*Ctnnb1*^ex3fl/wt mice (Supplementary Fig. 4i–l). The downstream effectors of IL-22, such as anti-microbial peptides RegIIIβ and RegIIIγ, were also dramatically diminished post-activation of β-catenin signaling in ILC3s (Supplementary Fig. 4m), indicating that activated Wnt/β-catenin signaling in ILC3s impairs host defense and promotes the development of infectious colitis.

Collectively, these data suggest that the precise regulation of Wnt/β-catenin signaling is crucial for intestinal homeostasis, and activation of Wnt/β-catenin signaling in ILC3s increases susceptibility to colitis in mice and may contribute to IBD in humans.

### The activation of Wnt/β-catenin signaling influences cell survival and proliferation in ILC3s

Because of few ILC3s in *Rorc*^cre*Ctnnb1*^ex3fl/wt mice, to further investigate the mechanism by which Wnt/β-catenin pathway regulating ILC3s, *Ctnnb1*^ex3fl/fl mice were crossed with *CreER*^T2 transgenic mice. Then ILC3s were isolated from *CreER*^T2*Ctnnb1*^ex3fl/wt mice and treated with 4-hydroxytamoxifen, which induces Cre-mediated DNA recombination to remove *Ctnnb1* exon 3 in ILC3s. As shown in Fig. 4a, b, ILC3s derived from *CreER*^T2*Ctnnb1*^ex3fl/wt mice gradually exhibited increased β-catenin expression but reduced RORγt expression post 4-hydroxytamoxifen treatment. Compared with the significantly increased ILC3s numbers from the control group, the number of ILC3s with activated β-catenin signaling was not increased after being cultured for 9 days in vitro (Fig. 4c). Furthermore, *CreER*^T2*Ctnnb1*^ex3fl/wt ILC3s exhibited more cell death and fewer Ki-67⁺ cells (Fig. 4d–g). To further verify whether activated β-catenin signaling regulated ILC3s proliferation, ILC3s were labeled with CFSE and then cultured with 4-hydroxytamoxifen in vitro. Compared with *Ctnnb1*^ex3fl/wt ILC3s, ILC3s from *CreER*^T2*Ctnnb1*^ex3fl/wt mice showed less proliferative ability (Fig. 4h, i). Interestingly, even in *CreER*^T2*Ctnnb1*^ex3fl/wt ILC3s group, the β-catenin⁺ subset showed less proliferation than the β-catenin⁻ subset (Fig. 4h, i), suggesting that Wnt/β-catenin signaling intrinsically regulates ILC3s proliferation. To determine whether Wnt/β-catenin signaling has different effects on ILC3 subsets, CCR6⁺, Nkp46⁺, and CCR6/Nkp46 double negative (DN) ILC3s were sorted by flow cytometry, then were treated with 4-hydroxytamoxifen and evaluated with cell viability and proliferation capability. After β-catenin activation, every subset of ILC3s showed an increase in cell death and a decrease in proliferation (Supplementary Fig. 5a–d), similar to the general ILC3 response. Additionally, the same number of *CreER*^T2*Ctnnb1*^ex3fl/wt ILC3s and control ILC3s were transferred into *Rag2*^−/− *Il2rg*^−/− recipients and then treated with tamoxifen to activate the Wnt pathway in ILC3s in vivo. Nearly half of *CreER*^T2*Ctnnb1*^ex3fl/wt ILC3s were positive for β-catenin⁺ at day 6 (Supplementary Fig. 5e). There are fewer *CreER*^T2*Ctnnb1*^ex3fl/wt ILC3s than control ILC3s (Supplementary Fig. 5f), which was consistent with the results obtained in vitro. In addition, activation of the Wnt pathway in common helper ILC progenitors (CHILPs) also inhibited the development or maintenance of ILC3s (Supplementary Fig. 5g–i). Collectively, Wnt pathway activation may influence ILC3s through inhibiting cell proliferation, limiting cell development from progenitor cells, and promoting cell death. Next, to further understand the effect of activated Wnt/β-catenin signaling on ILC3s, the isolated ILC3s were treated with the Wnt activator and examined with RNA sequencing (RNA-seq). ILC3s with or without activated Wnt/β-catenin signaling displayed different gene expression profiles based on PCA analysis

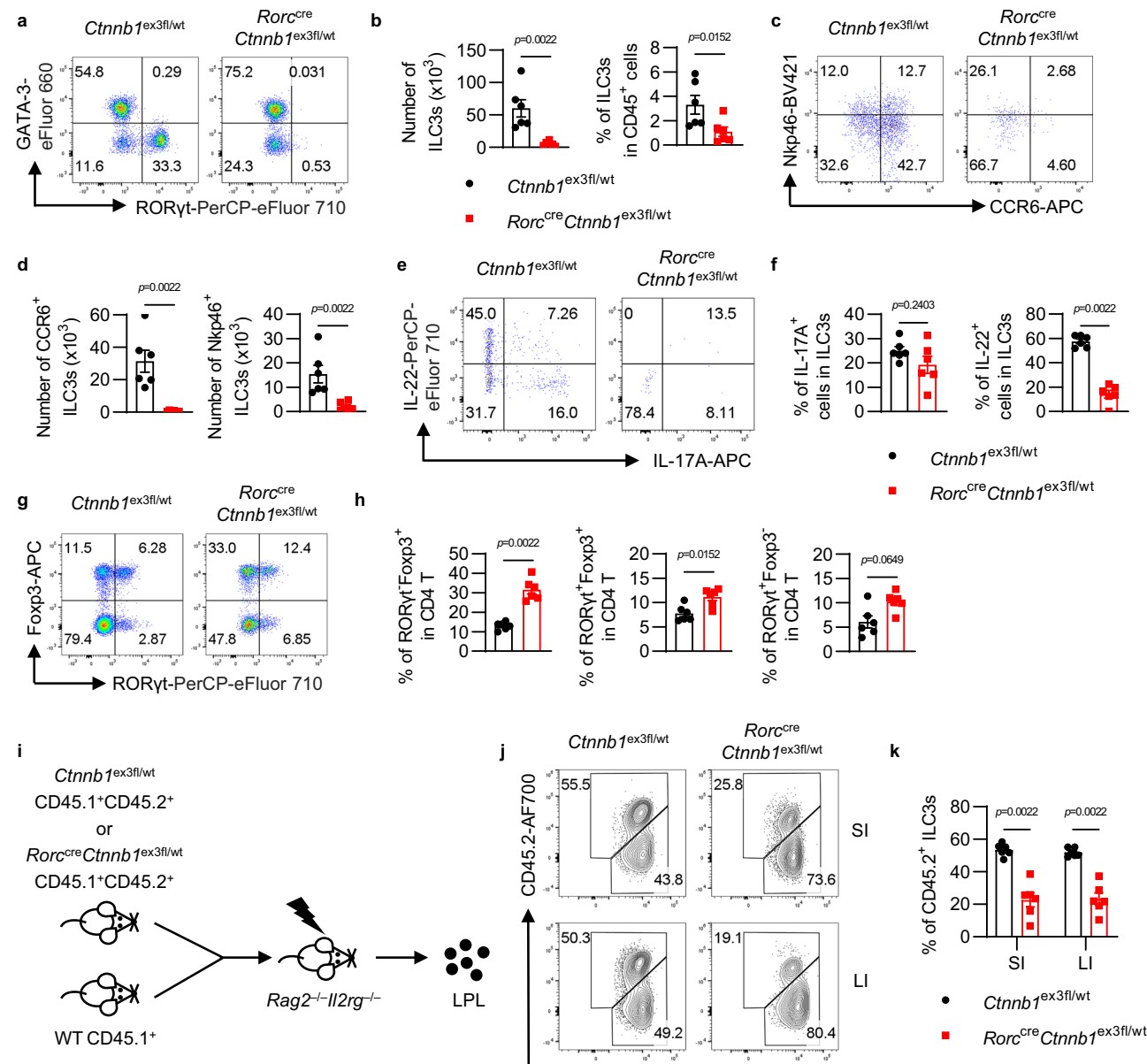

**Fig. 2 | Dysregulation of β-catenin leads to ILC3s' deficiency in vivo. a–h** The *Ctnnb1*^ex3fl/wt and *Rorc*^cre*Ctnnb1*^ex3fl/wt mice were euthanized at 8-week-old and the colonic lamina propria lymphocytes (LPLs) were analyzed. **a** Flow cytometry analysis of ILCs (gated in lineage⁻CD127⁺). **b** The number and percentage of ILC3s in total CD45⁺ cells. **c** Flow cytometry analysis of ILC3 subsets (gated in lineage⁻CD127⁺RORγt⁺). **d** The number of CCR6⁺ and Nkp46⁺ ILC3s. **e** Flow cytometry analysis of cytokine production in ILC3s (gated in lineage⁻CD127⁺RORγt⁺). Cytokine production was detected under PMA/Ionomycin treatment. **f** The percentage of IL-17A⁺ and IL-22⁺ cells in total ILC3s. **g** Flow cytometry analysis of CD4⁺ T cells (gated in CD3⁺CD4⁺). **h** The percentage of RORγt⁻Foxp3⁺, RORγt⁺Foxp3⁺ and RORγt⁺Foxp3⁻ cells in total CD4⁺ T cells. **i–k** Bone marrow cells isolated from wild

type (CD45.1⁺) and *Ctnnb1*^ex3fl/wt (CD45.1⁺CD45.2⁺) or *Rorc*^cre*Ctnnb1*^ex3fl/wt (CD45.1⁺CD45.2⁺) were mixed at a 1:1 ratio and transferred to irradiated *Rag2*⁻/⁻*Il2rgc*⁻/⁻ recipients. **i** Experimental design for ILC3 competition assay in mixed bone marrow (BM) chimera mice. **j** Flow cytometry analysis of CD45.1⁺ and CD45.1⁺CD45.2⁺ ILC3s (gated in lineage⁻CD127⁺RORγt⁺) in small intestine (SI) and large intestine (LI) in BM chimera mice. **k** The percentage of CD45.2⁺ ILC3s (gated in lineage⁻CD127⁺RORγt⁺CD45.1⁺CD45.2⁺) in total ILC3s from the small intestine (SI) and large intestine (LI). Each dot represents one individual mouse (*n* = 6 per group). Error bars represent the SEM. Statistical significance was tested by unpaired two-sided Mann–Whitney *U* test. Data are representative of three independent experiments.

(Supplementary Fig. 6a, b). Most ILC3-related transcription factors were suppressed by the activation of Wnt/β-catenin signaling, such as *Rorc*, *Stat3*, *Gata3*, *Id2*, and *Rara* (Fig. 4j), suggesting that ILC3 maintenance requires precise controlled Wnt/β-catenin signaling. Consistent with the impairment of ILC3s function, the expression of most effector molecules in ILC3s, including surface protein *Ltb*, *H2-Ab1*, *H2-K1*, and cytokines *Il22*, *Il2*, *Csf2*, *Vegfa*, decreased upon β-catenin dysregulation. Interestingly, IL-2 receptor *Il2rb* and *Il2rg* expression was also significantly decreased in β-catenin-activated ILC3s, and this might contribute to the reduced proliferation of β-catenin-activated

ILC3s, as IL-2 and IL-7 signaling are crucial for ILC3 proliferation[23–25]. Gene Ontology (GO) analysis further revealed that cell proliferation-promoting associated pathways (including positive regulation of cell cycle, DNA replication initiation, regulation of cell division, lymphocyte proliferation) were remarkably decreased in β-catenin-activated ILC3s, while cell growth-inhibiting or cell death associated pathways (including neuron apoptotic process, negative regulation of cell growth, positive regulation of autophagy) were enriched in ILC3s after Wnt activator treatment (Fig. 4k and Supplementary Fig. 6c–g), indicating that Wnt/β-catenin signaling was indeed

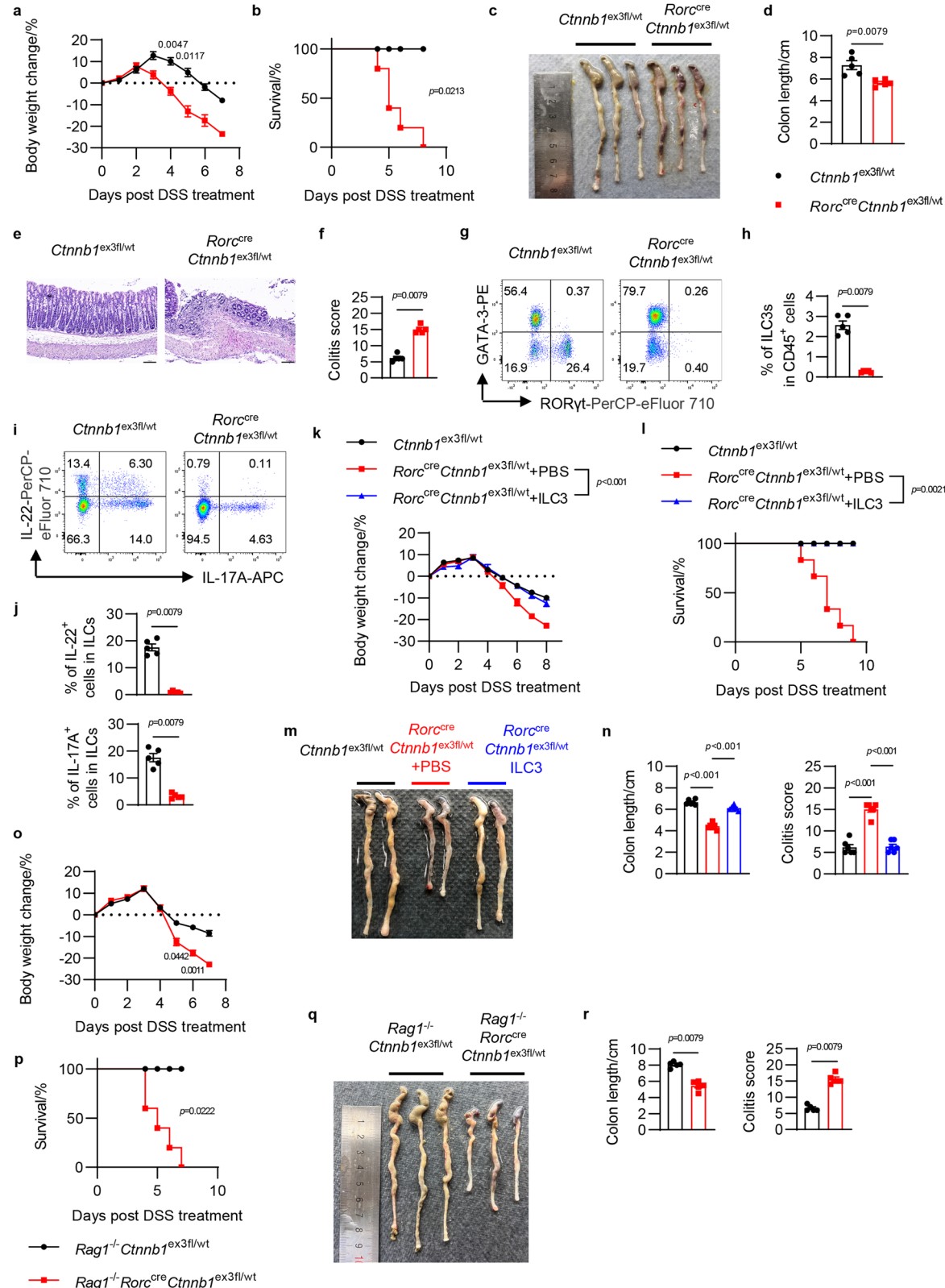

involved in the regulation of ILC3s survival and proliferation. Additionally, immune regulation pathways in ILC3s (including negative regulation of immune system process and regulation of inflammatory response) were also modified by Wnt/β-catenin signaling activation (Fig. 4k), which was consistent with the dysfunction of ILC3s observed in IBD and CRC patients. Overall, these data indicate that dysregulation of the Wnt/β-catenin pathway may reduce ILC3s number by promoting cell death and inhibiting proliferation.

## TCF-1 could not directly inhibit RORγt expression in β-catenin-activated ILC3s

RORγt is the master transcription factor for ILC3s, whose deficiency results in ILC3 deficiency. Thus, we next aimed to understand how

**Fig. 3 | Activated Wnt/β-catenin signaling in ILC3 increases susceptibility to colitis. a–j** 8-week-old *Rorc*^cre*Ctnnb1*^ex3fl/wt mice (*n* = 5) and their littermates *Ctnnb1*^ex3fl/wt mice (*n* = 5) were given 3% DSS in drinking water for 5 days. **a, b** Body weight change (**a**) and survival rates (**b**) at different time points. **c, d** Gross morphological changes (**c**) and lengths of colons (**d**). **e** Histological analysis of representative colons. Scale bars represent 100 µm. **f** Colitis score of *Ctnnb1*^ex3fl/wt and *Rorc*^cre*Ctnnb1*^ex3fl/wt mice. **g** Flow cytometry analysis of colonic ILCs (gated in lineage⁻CD127⁺). **h** The percentage of colonic ILC3s in total CD45⁺ cells. **i** Flow cytometry analysis of cytokine production in colonic ILCs (gated in lineage⁻CD127⁺). Cytokine production was detected under PMA/Ionomycin treatment. **j** The percentage of IL-17A⁺ and IL-22⁺ cells in total colonic ILCs. **k–n** 8-week-old *Rorc*^cre*Ctnnb1*^ex3fl/wt mice and their littermates *Ctnnb1*^ex3fl/wt mice were given 2.5% DSS in drinking water for 5 days, followed by normal drinking water. Mice were divided into three groups: *Ctnnb1*^ex3fl/wt mice (*n* = 8), *Rorc*^cre*Ctnnb1*^ex3fl/wt mice with or without ILC3s replenishment (*n* = 6 per group). **k–l** Body weight change (**k**) and survival rates (**l**) at different time points post DSS treatment. **m** Gross morphological changes of colons from three groups. **n** Lengths of colons and colitis score from three groups (*n* = 6 per group). **o–r** 8-week-old *Rag1*⁻/⁻*Rorc*^cre*Ctnnb1*^ex3fl/wt mice (*n* = 5) and their littermates *Rag1*⁻/⁻*Ctnnb1*^ex3fl/wt mice (*n* = 5) were given 3% DSS in drinking water for 5 days. **o, p** Body weight change (**o**) and survival rates (**p**) at different time points post DSS treatment. **q** Gross morphological changes of colons. **r** Lengths of colons and colitis score. Each dot represents one individual mouse. Error bars represent the SEM. Statistical significance was tested by two-sided two-way ANOVA with Sidak correction adjusted for multiple comparisons (**a, k, o**), two-sided log-rank Mantel–Cox test (**b, l, p**), two-sided one-way ANOVA with Tukey's test adjusted for multiple comparisons (**n**) and unpaired two-sided Mann–Whitney *U* test (**d, f, h, j, r**). Data are representative of two (**k–r**) or three (**a–l**) independent experiments.

activated β-catenin signaling inhibited RORγt expression in ILC3s. Upon activation, β-catenin accumulates in the cytoplasm, eventually translocates to the nucleus, and acts as a transcriptional coactivator to interact with TCF/LEF transcription factors. This β-catenin/TCF-1 complex has been shown to bind to the promoter of downstream genes, including *Rorc*, and regulate T cell differentiation and function[26,27]. As expected, the upregulation of interaction between β-catenin and TCF-1 was also detected in β-catenin-activated ILC3s by co-immunoprecipitation assay (Fig. 5a). Furthermore, ILC3s isolated from *Ctnnb1*^ex3fl/wt (Ctrl) and *CreER*^T2*Ctnnb1*^ex3fl/wt (KO) mice were treated with 4-hydroxytamoxifen, then subjected to Cleavage Under Targets and Tagmentation (CUT&Tag) assays with anti-TCF-1 antibody. Results showed that TCF-1 could bind to three regions at *Rorc* locus (R1, R2, R3) in ILC3s (Fig. 5b). Compared with control ILC3s, KO ILC3s had more TCF-1 binding at R1 region, which shared overlapping sequence with *Rorc* promoter. As shown by Luciferase assay in 293T cells, increased TCF-1 binding to R1 region promoted downstream gene expression (Fig. 5c). Although this result is consistent to previous reports that ectopic expression of β-catenin/TCF-1 could promote the expression of *Rorc* in 293T and T cells by direct binding[21], it is contrary to the downregulation of RORγt by activated β-catenin/TCF-1 signaling in ILC3s. Thus, these data suggest that RORγt expression in ILC3s is not directly inhibited by TCF-1 binding to the *Rorc* promoter (R1).

### Activated β-catenin signaling influences ILC3s via epigenetic regulation mechanisms

To further investigate how Wnt/β-catenin pathway regulates ILC3s, ILC3s isolated from *Ctnnb1*^ex3fl/wt (Ctrl) and *CreER*^T2*Ctnnb1*^ex3fl/wt (KO) mice were treated with 4-hydroxytamoxifen, followed by histone modification (H3K4me3 and H3K27ac) analysis via CUT&Tag assays and deep sequencing. Interestingly, most gene loci in KO cells exhibited reduced H3K4me3 and H3K27ac modifications (Fig. 5d). Consistent with the RNA-seq data (Fig. 4i), the genes that expression was reduced in Wnt/β-catenin-activated ILC3s, such as *Rorc, Junb, Nfatc2, Il22, Il7, Vegfa, Il2rb* and genes involved in regulating cell proliferation or chemotaxis, were also associated with reduced H3K4me3 and H3K27ac modifications (Fig. 5d, e), indicating that activated β-catenin signaling may influence ILC3s through epigenetic regulation mechanisms. To further determine the epigenetic regulation of Wnt/β-catenin pathway in ILC3s, we further performed Assay for Transposase Accessible Chromatin with high-throughput sequencing (ATAC-seq) with control and Wnt-activated ILC3s. As shown in Fig. 5f, the ILC3-related key transcription factors RORγt and RORα showed more accessible chromatin sites in control ILC3s, while TCFs showed more accessible chromatin sites in Wnt-activated ILC3s, which was consistent with the decreased RORγt and increased TCF1 signaling post-Wnt/β-catenin activation in ILC3s (Fig. 5a). Together, these data demonstrate that activated Wnt/β-catenin pathway may influence ILC3s through regulating the epigenetic modification and chromatin accessibility of the crucial genes in ILC3s.

### NFATc2 and JunB regulate RORγt expression in ILC3s

To further identify how Wnt/β-catenin signaling regulates RORγt expression, we examined chromatin-accessible regions in the *Rorc* locus which also had pol II binding and histone marks including H3K4me3 and H3K27ac. A total of six different chromatin-accessible sites (named P1, P2, P3, P4, P5, and P6) in the *Rorc* locus were found (Fig. 6a). Less chromatin accessibility, pol II binding, and H3K4me3 and H3K27ac modifications were observed at these six regions in KO ILC3s than control ILC3s, indicating diminished transcription activity at *Rorc* locus caused by β-catenin activation. Since P1–P4 sites are near the RORc promoter, where multiple transcription factors (TFs) bind to regulate gene transcription, these results also suggest that the activation of Wnt/β-catenin signaling may regulate RORγt expression by influencing the activities of these TFs.

Next, the potential TFs binding with P1–P4 regions were predicted with MotifFinder. Based on the differentially expressed genes (log₂FC > 1 or log₂FC < −1) in RNA-seq data, 11 potential TFs regulated by β-catenin signaling at those four regions were proposed (Fig. 6b, c). Further, quantitative PCR was used to verify the difference in expression of those TFs, and five of them, including *Nfatc2, Junb, Klf7, Foxs1* and *Irf7*, showed similar changes to those found in RNA-seq (Supplementary Fig. 7a). To determine whether these five TFs could regulate *Rorc* promoter activity, luciferase reporter systems with P1, P2, P3 or P4 regions were constructed, respectively. NFATc2, KLF7, and JunB were capable of promoting downstream gene expression after binding to *Rorc* promoter P3 region (Fig. 6d), while FOXS1 and IRF7 showed no effect on *Rorc* transcription activity in luciferase assay (Supplementary Fig. 7b). Nevertheless, the expression of *Klf7* is increased and the expression of *Nfatc2* and *Junb* is decreased in β-catenin-activated ILC3s (Supplementary Fig. 7a), suggesting that NFATc2 and JunB are able to directly regulate RORγt expression in ILC3s.

To determine whether Wnt/β-catenin signaling regulates RORγt expression through NFATc2 and JunB in ILC3s, the CUT&Tag assays with anti-NFATc2 or anti-JunB antibodies were performed in control and KO ILC3s. Results showed that both NFATc2 and JunB could bind to P3, P5, and P6 sites within the *Rorc* locus (Fig. 6e), and less NFATc2 and JunB binding was observed at the P3 site in β-catenin-activated ILC3s than control ILC3s. Moreover, in consistent with the decreased RNA levels of NFATc2 and JunB (Supplementary Fig. 7a), their proteins were also significantly decreased in β-catenin-activated ILC3s shown by western blot assay (Fig. 6f). These data indicate that the activation of Wnt/β-catenin signaling suppresses the expression of NFATc2 and JunB in ILC3s, which further affects their transcription activities on *Rorc* expression. To further demonstrate the regulation effect of NFATc2 and JunB on RORγt expression in ILC3s, small interfering RNA (siRNA) against *Nfatc2* or *Junb* were transfected into ILC3s and partially inhibited their mRNA expression (Supplementary Fig. 7c, d). By targeting *Nfatc2* and *Junb* with siRNA, the RORγt expression in ILC3s was significantly downregulated at both protein and mRNA levels (Fig. 6g, h). Together, these data suggest that dysregulated Wnt/β-catenin

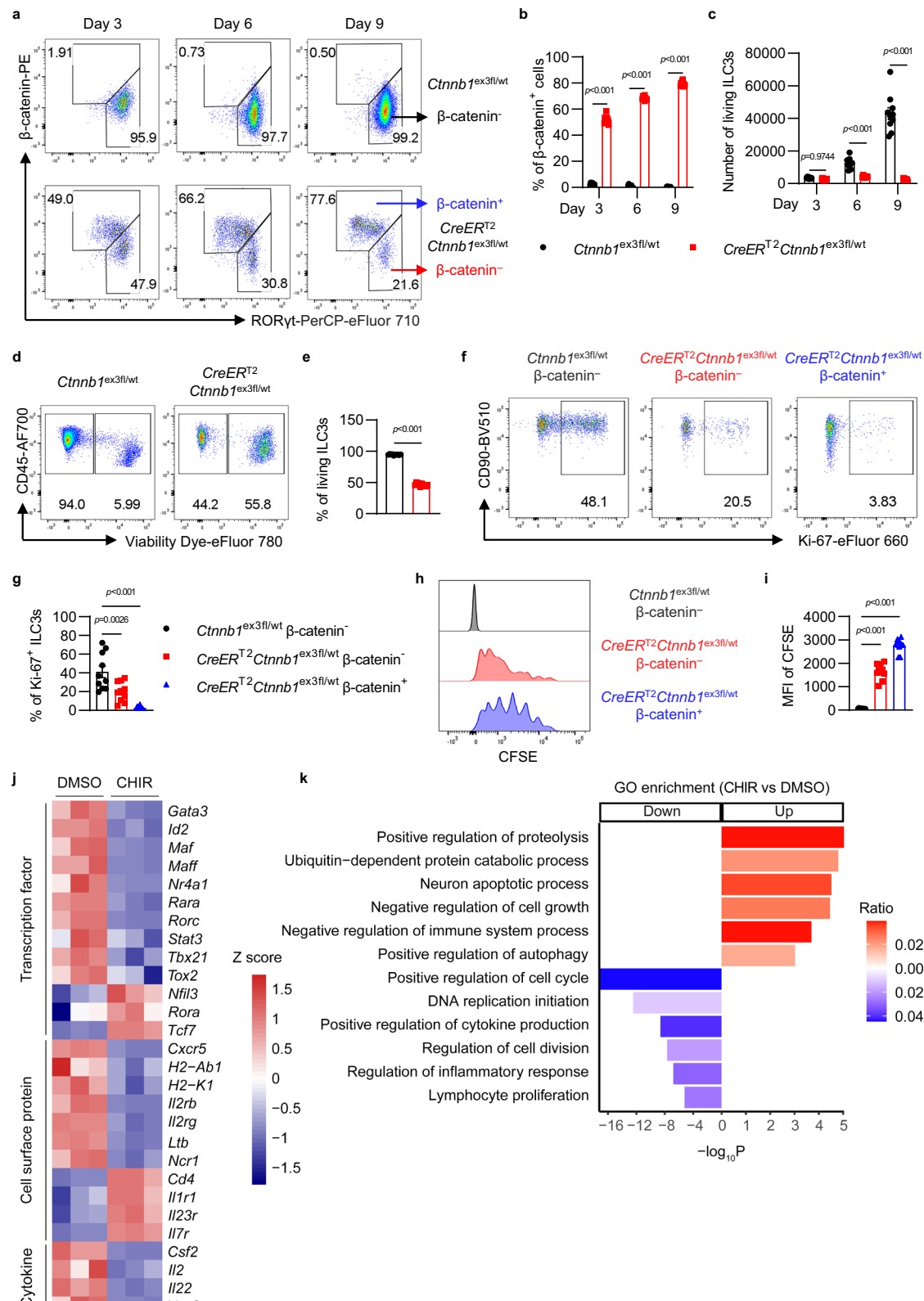

signaling may suppress RORγt expression by inhibiting NFATc2 and JunB in ILC3s.

## TCF-1 binds to *Junb* promoter and suppresses JunB expression in ILC3s

Next, to determine how Wnt/β-catenin signaling regulates NFATc2 and JunB expression, the chromatin accessibility, H3K4me3, and H3K27ac modifications, and the binding of TCF-1 at *Nfatc2* and *Junb* regions were examined with ATAC-seq and CUT&Tag assays. Although there was no difference in chromatin accessibility between Ctrl and KO ILC3s, the histone marks H3K4me3 and H3K27ac showed a significant reduction at *Nfatc2* and *Junb* regions (Fig. 7a, b), suggesting that activation of β-catenin signaling inhibited the transcription activities of *Nfatc2* and *Junb*. Interestingly, more TCF-1 binding was found at *Nfatc2*

**Fig. 4 | The activation of Wnt/β-catenin signaling influences cell survival and proliferation of ILC3s. a–i** ILC3s (gated in 7AAD⁻lineage⁻CD90^high^CD45^low^) from *Ctnnb1*^ex3fl/wt^ mice and *CreER*^T2^*Ctnnb1*^ex3fl/wt^ mice were isolated and cultured in vitro with OP9 feeder cells. ILC3s were treated with 4-hydroxytamoxifen at day 0. Cells from *CreER*^T2^*Ctnnb1*^ex3fl/wt^ mice were divided into two subsets: β-catenin⁻ ILC3s and β-catenin⁺ ILC3s. **a** Flow cytometry analysis of RORγt and β-catenin expression at day 3, day 6, and day 9. **b** The percentage of β-catenin⁺ ILC3s at day 3, day 6 and day 9. **c** The number of total living ILC3s on day 3, day 6, and day 9. **d** Flow cytometry analysis of cell death at day 9. **e** The percentage of living ILC3s at day 9. **f** Flow cytometry analysis of Ki-67⁺ ILC3s in *Ctnnb1*^ex3fl/wt^ β-catenin⁻ ILC3s, *CreER*^T2^*Ctnnb1*^ex3fl/wt^ β-catenin⁻ ILC3s and *CreER*^T2^*Ctnnb1*^ex3fl/wt^ β-catenin⁺ ILC3s at day 9. **g** The percentage of Ki-67⁺ ILC3s in different groups at day 9. **h** Representative flow cytometric histograms of CFSE-labeled ILC3s in different groups at day 9. **i** Mean fluorescence intensity (MFI) of CFSE in different groups

at day 9. **j, k** Living ILC3s (gated in 7AAD⁻lineage⁻CD90^high^CD45^low^) from *Rag1*⁻ᐟ⁻ mice were treated with DMSO and CHIR-99021 (CHIR) for 24 hours in vitro. The mRNA of DMSO-treated and CHIR-treated ILC3s was isolated, and then RNA-seq was performed. **j** Heatmap of normalized counts comparing gene expression for transcripts related to ILC3's development and function. Scale based on *Z* score of log₂(normalized counts). **k** GO enrichment analysis of differentially expressed genes based on RNA-seq data. Red represents pathways enriched by upregulating genes in CHIR-treated ILC3s, and blue represents pathways enriched by down-regulating genes in CHIR-treated ILC3s. Each dot represents one individual replicate (*n* = 10 per group in **b, c, e, g, i**). Error bars represent the SEM. Statistical significance was tested by unpaired two-sided Student's *t*-test (**b, c, e**), two-sided one-way ANOVA with Tukey's test adjusted for multiple comparisons (**g, i**) and one-sided hypergeometric test adjusted by the Benjamini−Hochberg (BH) procedure (**k**). Data are representative of three independent experiments.

introns (N1) and *Junb* promoter (J1) in β-catenin-activated ILC3s (Fig. 7a, b). Luciferase reporter assay was further conducted to examine the direct effect of TCF-1 on the *Nfatc2* N1 site and *Junb* J1 site. Interestingly, TCF-1 overexpression showed mild influence on the expression of the gene with *Nfatc2* N1 site but dramatically suppressed the expression of a downstream gene with *Junb* J1 site (Fig. 7c, d), indicating that more TCF-1 accumulation after β-catenin activation could directly suppress JunB expression in ILC3s.

Altogether, these data reveal that dysregulation of Wnt/β-catenin signaling in ILC3s may inhibit RORγt expression by regulating JunB, leading to ILC3s deficiency and aggravated intestinal inflammation.

## Discussion
ILC3s have shown a broadly protective role in various murine colitis and CRC models[9,28]. IL-22 produced by ILC3s is a key component in maintaining intestinal homeostasis through promoting epithelium integrity and regulating microbiota[12,29]. MHC-II-expressing ILC3s exhibit the ability to present antigens to CD4⁺ T cells, inhibiting microbiota-specific T cell responses and preventing gut inflammation[30,31]. Specific deletion of MHC-II⁺ ILC3 in mice also promotes CRC tumor development[18]. ILC3s are also required for intestinal Treg development via expressing IL-2, MHC-II, and GM-CSF[30,32,33]. In line with these discoveries in mice, intestinal ILC3s in patients with IBD or CRC are dramatically reduced or exhibit diminished expression of MHC-II and IL-2 compared with healthy individuals[18,32]. However, the underlying mechanism is not clear. In our study, we identified an aberrant activation of Wnt/β-catenin signaling in ILC3s from patients with IBD and CRC. Interestingly, a recent study also suggested that increased *CTNNB1* expression by ILCs in pediatric-onset colitis[34]. Wnt/β-catenin signaling, a well-known pathway implicated in the development of IBD and CRC, is widely activated in the inflammatory and tumor microenvironment of the intestine. ILC3s, as a critical immune regulator in the gut microenvironment, are also remarkably influenced by the dysregulated Wnt/β-catenin signaling. We found that the activation of Wnt/β-catenin signaling not only leads to a dramatic reduction of ILC3s but also promotes the dysfunction of ILC3s with reduced expression of IL-22, MHC-II, IL-2, GM-CSF, which further aggravates the DSS-induced colitis in mice. Whether blocking the dysregulated Wnt/β-catenin signaling in ILC3s or promoting the ILC3s functional normalization could alleviate IBD and CRC in patients requires further investigation.

Wnt signaling plays a pivotal role in regulating homeostatic self-renewal in various adult tissues, typically acting as a positive regulator of cell maintenance and proliferation in stem cells and epithelial cells[1,35]. Aberrant activated Wnt/β-catenin signaling is widely implicated in numerous malignancies, including cancers of the gastrointestinal tract[35]. Wnt signaling is also required for T-cell development in the thymus and might also be involved in developing B cells in the bone marrow[21,36]. And activation of Wnt signaling contributes to the survival of regulatory T cells[36]. Conversely, activation of Wnt/β-catenin signaling dramatically inhibits the survival and proliferation of ILC3s,

highlighting the necessity for further investigation into the distinct regulatory mechanisms involved. Additionally, RORγt, the master transcriptional factor for ILC3s development and function, also plays a critical role in T cell development and differentiation. Activation of Wnt/β-catenin signaling also exhibits two distinct effects on RORγt expression in ILC3s and T cells: promoting RORγt expression in DP thymocytes, Th17, and Treg cells, while suppressing RORγt expression in ILC3s. Previous study shows that STAT3 also exhibits different regulation effects on RORγt expression between T cells and ILC3s[37]. Although ILC3s and Th17 cells are commonly considered counterparts in innate and adaptive immune systems, our data indicate that their regulatory mechanisms are more distinct and diversified than previously recognized. This insight emphasizes the need for a comprehensive examination of the unique regulatory pathways governing ILC3s in the context of Wnt/β-catenin signaling.

TCF-1, a central player in the canonical Wnt pathway, has been established as a key regulator of target genes[26]. Previous studies reveal the indispensable role of TCF-1 in ILCs' development and function[38–40]. Whereas, our results show that TCF-1 upregulation could impair ILC3's development and function. Whether less Wnt/β-catenin signaling is required for ILC3s needs to be further determined with ILC3-specific *Ctnnb1* deficient mice. Additionally, in mature ILC3, activation of the Wnt pathway results in RORγt downregulation and cell death. It is possible, however, that the activated Wnt pathway upregulates TCF-1 expression and prevents stem-like ILC progenitors from undergoing differentiation into ILC3. Future studies should utilize TCF-1 reporter mice in order to examine those TCF-1^high^ stem-like progenitor cells. The molecular regulatory network between TCF-1 and RORγt is complicated as we find that direct binding of TCF-1 to *Rorc* stimulates gene expression in 293T cells in vitro. It is possible that TCF-1 plays different roles in regulating RORγt expression in different cell types. There may be a variety of co-factors that affect TCF-1 transcription activity, which may be expressed differently in different cell types. A most recent paper reported that TCF-1 is essential for the development of non-LTi ILC subsets including NCR⁺ ILC3s[41], possibly by regulating the ILC progenitor. While TCF-1 is not required for LTi development, they found that it is able to promote LT expression in LTi cells for Peyer's patch formation. Interestingly, their data also indicated that TCF-1 deficiency in LTi cells might lead to an increased RORγt expression, which is in agreement with our finding that TCF-1 over-activation results in decreased RORγt expression in ILC3s, indicating that TCF-1 may have a dose-dependent and stage-dependent effect on RORγt expression and ILC3 development and function. Recent studies reveal more functions of TCF-1 in T cells, such as the function of HDAC to regulate gene expression and participate in controlling global genome organization[42–44]. Our epigenetic analysis, encompassing ATAC-seq, pol II binding sites, and histone modification, indicates that the *Rorc* locus in Wnt-activated ILC3s is less accessible and associated with reduced pol II binding and active epigenetic modifications. This suggests that activated Wnt/β-catenin signaling may modulate ILC3s

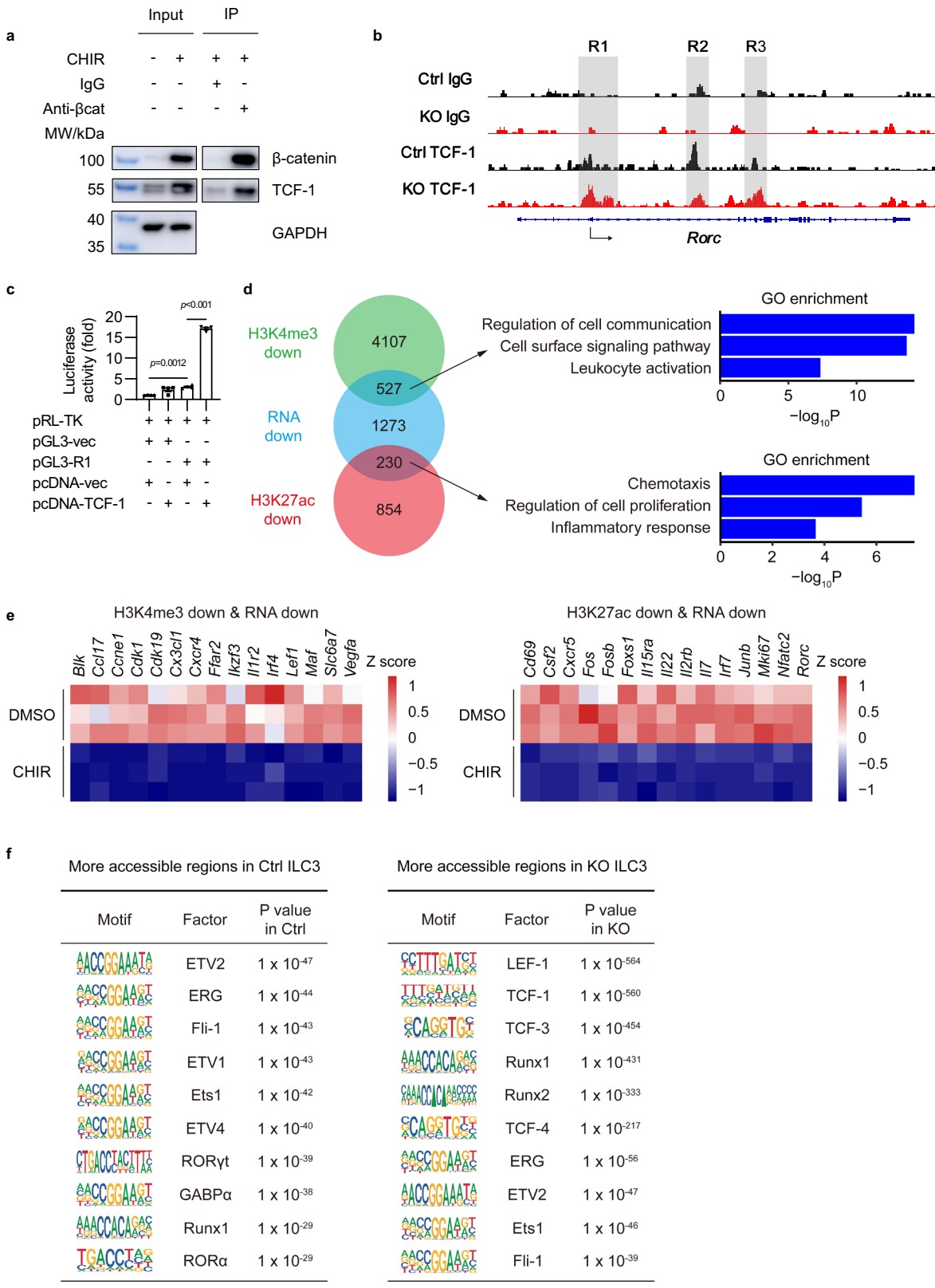

through epigenetic mechanisms. We further identified JunB and NFATc2 as the critical transcriptional factors directly regulating *Rorc* expression in ILC3s. The NFAT and AP-1 pathways are well-known TCR downstream pathways, but it is still not clear which signal could activate these two pathways in TCR-deficient ILC3s. JunB and NFATc2 can also regulate RORγt expression in T cells[45,46]. Our data suggest that AP-1 and NFAT pathways are specifically regulated and important for ILC3s, which emphasizes the need for further investigation, potentially utilizing JunB and NFATc2 deficient mice.

In addition, our study has several limitations that should be considered. Although we have demonstrated the potential role of dysfunctional Wnt/β-catenin pathway in ILC3s for intestinal diseases,

**Fig. 5 | Activated β-catenin signaling influences ILC3s via epigenetic regulation mechanisms. a** Living ILC3s isolated from *Rag1*[−/−] mice were treated with DMSO and CHIR-99021 (CHIR) for 24 hours in vitro. The immunoprecipitation (IP) was performed with IgG or anti-β-catenin antibody. Immunoblotting (IB) of β-catenin, TCF-1, and GAPDH was shown. **b** ILC3s from *Ctnnb1*[ex3fl/wt] (Ctrl) mice and *CreER*[T2]*Ctnnb1*[ex3fl/wt] (KO) mice were isolated and treated with 4-hydroxytamoxifen. The TCF-1 binding regions were detected by CUT&Tag assay and sequencing at day 9. Intergrative genomics viewer (IGV) browser view of control and TCF-1 binding peaks at *Rorc* gene locus is shown. **c** Luciferase activity in 293T cells transfected with vector alone (pGL3-vec) or vector containing *Rorc* promoter (pGL3-R1), together with empty vector (pcDNA-vec) or vector expressing TCF-1 (pcDNA-TCF-1). The results represent the relative firefly luciferase activities normalized to the corresponding Renilla luciferase activities. **d–f** ILC3s from *Ctnnb1*[ex3fl/wt] (Ctrl) and *CreER*[T2]*Ctnnb1*[ex3fl/wt] (KO) mice were treated with 4-hydroxytamoxifen. The histone modification marks and chromatin accessible regions were detected by CUT&Tag assay and sequencing on day 9. **d** Venn diagram comparing genes that gained less H3K4me3 modification in *CreER*[T2]*Ctnnb1*[ex3fl/wt] ILC3s (green), genes that gained less H3K27ac modification in *CreER*[T2]*Ctnnb1*[ex3fl/wt] ILC3s (red) and transcriptionally down regulated genes in CHIR-treated ILC3s (blue). GO enrichment analysis of Venn diagram overlapping genes is also shown. **e** RNA expression heatmap of indicated Venn diagram overlapping genes (*n* = 3 biological replicates for each group) are shown. **f** De novo transcription factor−binding motif analysis (HOMER) of more accessible regions in Ctrl and KO ILC3s. The most highly significantly enriched motifs and corresponding *P* values are listed. Each dot represents one individual replicate (*n* = 4 in **c**). Error bars represent the SEM. Statistical significance was tested by two-sided one-way ANOVA with Tukey's test adjusted for multiple comparisons (**c**). Data are representative of three independent experiments (**a**, **c**).

there is no direct evidence showing that high levels of β-catenin in a physiological situation lead to a loss of ILC3s. Further experiments are needed to determine which physiological Wnt ligands or upstream molecules could activate β-catenin signaling in ILC3s in IBD or CRC diseases. Using a physiological β-catenin signaling agonist, we may uncover the pathological effect of dysfunctional Wnt/β-catenin pathway in ILC3s at physiological state both in vitro and in vivo. Additionally, although we have proved that strong activation of the Wnt/β-catenin pathway inhibits RORγt expression, we could not rule out the possibility that chronically activated Wnt/β-catenin pathway can antagonize RORγt activity and inhibit its downstream genes. It would be meaningful to distinguish the possible effect of the Wnt/β-catenin pathway on RORγt expression from the transcriptional activity of RORγt in the future.

Collectively, our results show that activated β-catenin signaling can result in a deficiency of ILC3s and aggravated intestinal inflammation in mice. Mechanically, activated Wnt/β-catenin in ILC3s inhibits RORγt expression through epigenetic regulation and TCF-1-JunB pathway. The results not only demonstrate a pathological role for dysregulated Wnt/β-catenin signaling in ILC3s for intestinal diseases but also reveal that ILC3s and T cells are regulated differently in the disease environment, suggesting potential new therapeutic targets for IBD and cancer patients.

# Methods
## Mice
*Rag1*[−/−] mice were purchased from Jackson Laboratory. CD45.1 mice were from Chen Dong (Tsinghua University, Beijing). *Rorc*[cre] mice[47] were from Dan R. Littman (New York University, NY). *Ctnnb1*[ex3flox/flox] mice were from Xuanming Yang (Shanghai Jiao Tong University, Shanghai). Crossing *Ctnnb1*[ex3flox/flox] mice with *Rorc*[cre] mice generated *Rorc*[cre]*Ctnnb1*[ex3flox/wt] mice. Crossing *Rorc*[cre]*Ctnnb1*[ex3flox/wt] mice with *Rag1*[−/−] mice generated *Rag1*[−/−]*Rorc*[cre]*Ctnnb1*[ex3flox/wt] mice. *CreER*[T2] (*Rosa26-CreER*[T2]) mice were from The Jackson Laboratory (JAX: 008463). *CreER*[T2] was inserted after *Rosa26* promoter. *CreER*[T2]*Ctnnb1*[ex3flox/wt] mice were generated by crossing *CreER*[T2] mice with *Ctnnb1*[ex3flox/flox] mice. All mice used in the experiments were around 8 weeks old on a C57BL/6 background. Both male and female mice were used in all experiments. The mice were maintained under specific pathogen-free (SPF) conditions at Tsinghua University. All studies were approved by the Animal Care and Use Committee of Tsinghua University.

## DSS-induced colitis model
Seven to eight-week-old mice were administrated with 3% (w/v) or 2.5% (w/v) DSS (MP Biomedicals; molecular weight: 36−50 kDa) in their drinking water for 5 consecutive days and then were switched to normal sterile water. DSS-treated mice were monitored daily for weight and other signs of disease. Where indicated, ILC3s were isolated and transferred to *Rorc*[cre]*Ctnnb1*[ex3flox/wt] mice by intravenous injection at day 0.

## *C. rodentium* infectious colitis model
Mice were orally gavaged with *C. rodentium* strain DBS100 (ATCC 51459; American Type Culture Collection). *C. rodentium* was prepared by culturing in LB broth overnight and bacterial concentration was determined by measuring the optical density at 600 nm (OD600). Mice were gavaged with $2 \times 10^9$ CFU in 200 mL PBS. Body weight was measured every 2 days. Fecal pellets were collected, weighed, homogenized and serially diluted in sterile PBS and plated on MacConkey agar plates to determine the CFU at indicated days. *C. rodentium* colonies were identified based on morphology after 18−24 h of incubation at 37 °C. The severity of colitis was scored by assessing the following parameters: mouse vitality (0−3), stool frequency (0−3), stool viscosity (0−3), rectal bleeding (0−3), colon wall thickness (0−3) and crypt damage (0−3), generating a maximum total score of 18.

## Construction of bone marrow chimeras
Bone marrow (BM) cells from WT CD45.1 mice, and *Ctnnb1*[ex3fl/wt] mice or *Rorc*[cre]*Ctnnb1*[ex3fl/wt] mice were mixed at a 1:1 ratio and intravenously transferred into lethally irradiated (5.5 Gy × 2) *Rag2*[−/−]*Il2rg*[−/−] recipients. The chimeras were used for experiments after a 6-week reconstitution.

## Isolation of intestinal lamina propria cells
Intestines were cut open longitudinally and cut into 1.5 cm pieces, washed in 1× PBS, and shaken in 1× HBSS containing 1 mM DTT (Sigma, D0632), 5 mM EDTA, 3% FBS, and 10 mM HEPES at 37 °C for 20 min for twice. After washing with 1× HBSS containing 10 mM HEPES, the tissues were then digested in RPMI 1640 medium containing 0.05% DNase I (Sigma, DN25) and 0.1 mg/ml Liberase (Roche, 05401020001) at 37 °C for 20−30 min. Then the digested tissues were homogenized by gentleMACS Dissociator (Miltenyi) and passed through a 70 μm cell strainer. Mononuclear cells were harvested from the interphase of an 80% and 40% Percoll gradient after a spin at 2500 rpm for 20 min at room temperature without break.

## ILC2 and ILC3 culture in vitro
OP9 stromal cells were cultured and irradiated in a cell culture plate in advance. ILC2s were isolated and purified from the gut of the donor mice, gated in 7AAD[−]lineage[−]CD127[+]KLRG1[+]. ILC3s were isolated and purified from the gut of the donor mice, gated in 7AAD[−]lineage[−]CD90[high]CD45[low]. Cells were stained with the following antibodies: 7AAD (BioLegend, 420404), CD127 (clone A7R34, BioLegend, 135023), KLRG1 (clone 2F1/KLRG1, BioLegend, 138408), CD90.2 (clone 30-H12, BioLegend, 105335), CD45 (clone 30-F11, BioLegend, 103128) and lineage cocktail including CD3 (clone 17A2, BioLegend, 100244), B220 (clone RA3-6B2, BioLegend, 103204), TCRβ (clone H57-597, BioLegend, 109204), TCRγ/δ (clone GL3, BioLegend, 118103),

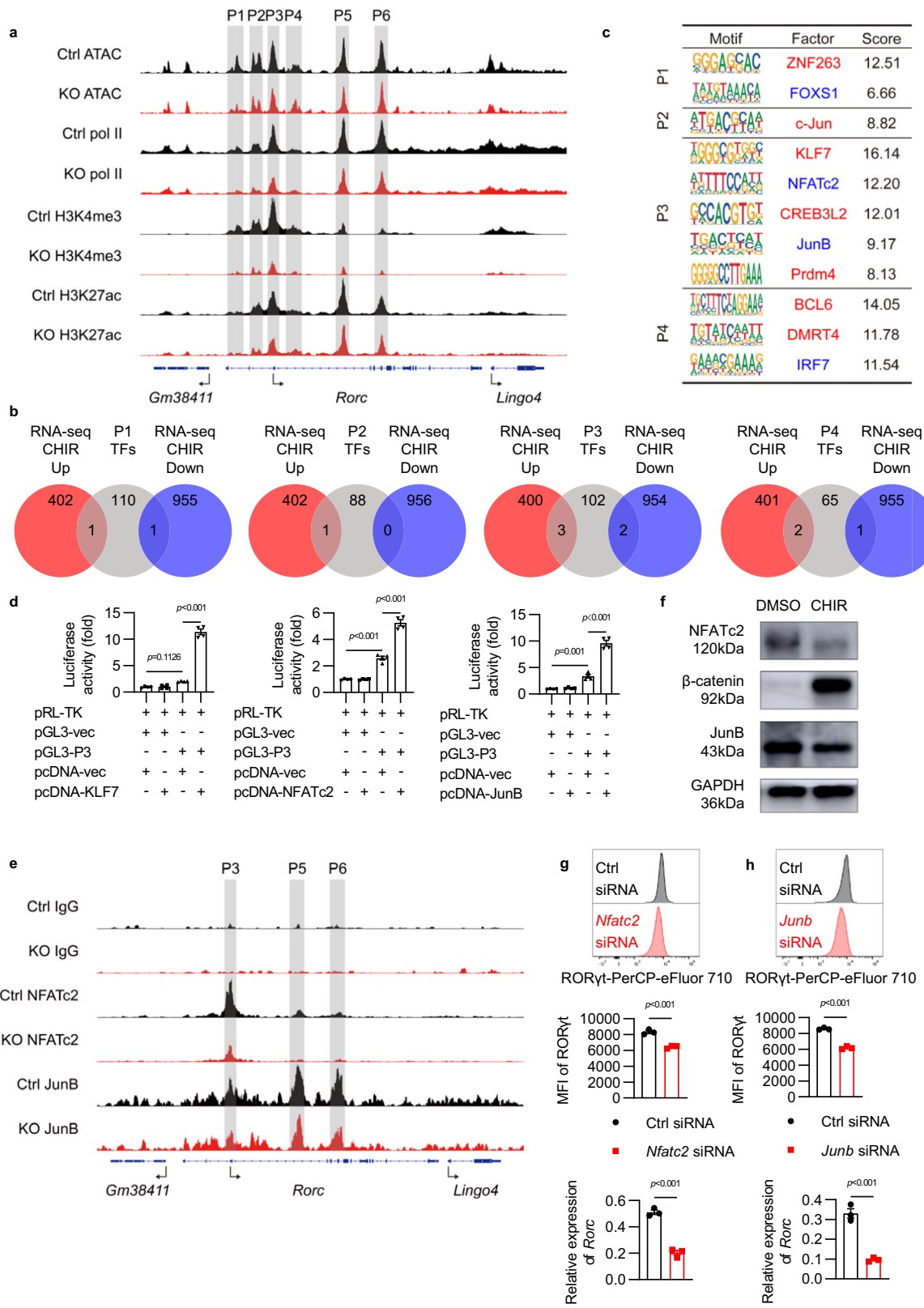

CD11b (clone M1/70, BioLegend, 101204), CD11c (clone N418, BioLegend, 117304), Ter-119 (clone TER-119, BioLegend, 116204), Gr-1 (clone RB6-8C5, BioLegend, 108404), NK1.1 (clone PK136, BioLegend, 108704). ILC2s were cultured with 10 ng/ml recombinant mouse IL-2 (Novoprotein, CK24) and 20 ng/ml recombinant mouse IL-7 (Novoprotein, CC73) in Alpha-MEM medium including 15% FBS, 10 mM HEPES, 100 U/ml Penicillin, 100 µg/ml Streptomycin, 1 mM sodium

pyruvate, 8 mg/ml Glutamine and 80 µM 2-Mercaptoethanol. ILC3s were cultured on OP9 cells (from Sheng Ding, Tsinghua University, Beijing) with 10 ng/ml recombinant mouse IL-2 (Novoprotein, CK24) and 20 ng/ml recombinant mouse IL-7 (Novoprotein, CC73) in Alpha-MEM medium including 15% FBS, 10 mM HEPES, 100 U/ml Penicillin, 100 µg/ml Streptomycin, 1 mM sodium pyruvate, 8 mg/ml Glutamine and 80 µM 2-Mercaptoethanol. DMSO, 20 µM CHIR-99021 (Selleck,

**Fig. 6 | NFATc2 and JunB regulate RORγt expression in ILC3s. a–c** ILC3s from *Ctnnb1*^ex3fl/wt (Ctrl) and *CreER*^T2*Ctnnb1*^ex3fl/wt (KO) mice were treated with 4-hydroxytamoxifen. The chromatin accessible regions, RNA polymerase II (pol II) binding regions, and histone modification marks (H3K4me3 and H3K27ac) were detected by CUT&Tag assay and sequencing. **a** Intergrative genomics viewer (IGV) browser view around *Rorc* gene locus is shown. P1–P6 indicate chromatin-accessible regions surrounding the *Rorc* gene locus. **b** Sequences of P1–P4 were selected and the potential binding transcription factors (TFs) were predicted by MotifFinder. Venn diagram comparing the predicted TFs which have the potential to bind to P1–P4 (gray), transcriptionally upregulated (red) and downregulated (blue) genes by RNA-seq in CHIR-treated ILC3s in Fig. 4. **c** The upregulated transcription factors (red) and downregulated transcription factors (blue) which have the potential to bind to P1–P4 are shown. **d** Luciferase activity in 293T cells transfected with vector alone (pGL3-vec) or vector containing P3 (pGL3-P3), together with empty vector (pcDNA-vec) or vector expressing KLF7 (pcDNA-KLF7), NFATc2

(pcDNA-NFATc2) and JunB (pcDNA-JunB), respectively. The results represent the relative firefly luciferase activities normalized to the corresponding Renilla luciferase activities. **e** ILC3s from *Ctnnb1*^ex3fl/wt (Ctrl) and *CreER*^T2*Ctnnb1*^ex3fl/wt (KO) mice were treated with 4-hydroxytamoxifen. The NFATc2 and JunB binding peaks at *Rorc* gene locus were detected by CUT&Tag assay and sequencing, as shown in the IGV browser view. **f** ILC3s isolated from *Rag1*^−/− mice were treated with DMSO and CHIR-99021 (CHIR) for 24 hours in vitro. Immunoblotting (IB) of NFATc2, β-catenin, JunB, and GAPDH are shown. **g, h** Flow cytometry analysis of RORγt expression and relative mRNA expression of *Rorc* in ILC3s transfected with Ctrl siRNA and *Nfatc2* siRNA (**g**) or *Junb* siRNA (**h**). Each dot represents one individual replicate (*n* = 4 in **d**, *n* = 3 in **g, h**). Error bars represent the SEM. Statistical significance was tested by two-sided one-way ANOVA with Tukey's test adjusted for multiple comparisons (**d**) and unpaired two-sided Student's *t*-test (**g, h**). Data are representative of three independent experiments (**d, f, g, h**).

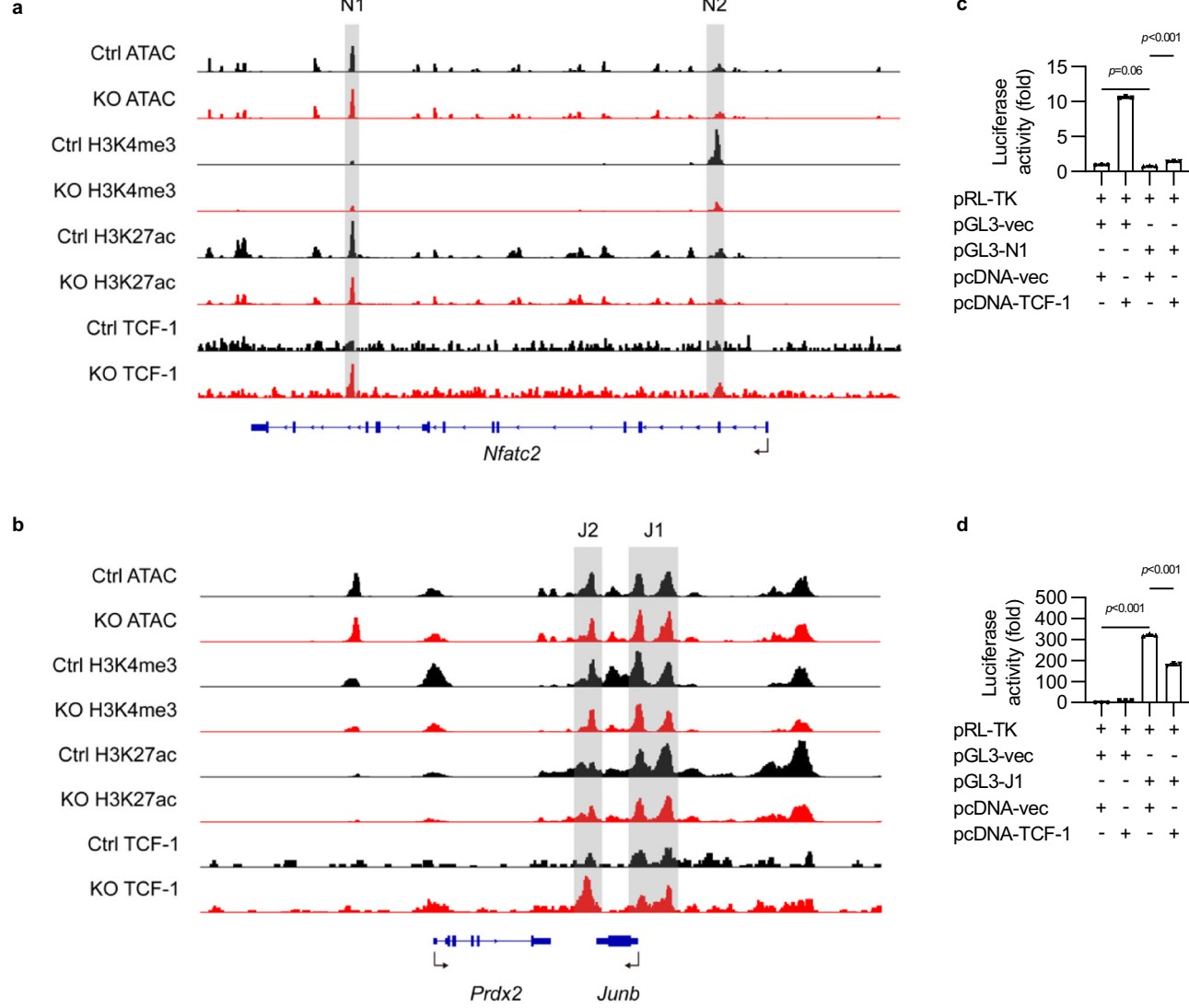

**Fig. 7 | TCF-1 binds to *Junb* promoter and suppresses JunB expression in ILC3s.**
**a, b** ILC3s from *Ctnnb1*^ex3fl/wt (Ctrl) and *CreER*^T2*Ctnnb1*^ex3fl/wt (KO) mice were treated with 4-hydroxytamoxifen. The chromatin accessible regions, histone modification marks, and TCF-1 binding peaks at *Nfatc2* (**a**) and *Junb* (**b**) gene locus were detected by ATAC-seq, CUT&Tag assay and sequencing, as shown in the IGV browser view. **c, d** Luciferase activity in 293T cells transfected with vector alone (pGL3-vec), vector containing *Nfatc2* locus N1 (pGL3-N1) (**c**) or vector containing *Junb* locus J1

(pGL3-J1) (**d**), together with empty vector (pcDNA-vec) or vector expressing TCF-1 (pcDNA-TCF-1). The results represent the relative firefly luciferase activities normalized to the corresponding Renilla luciferase activities. Each dot represents one individual replicate (*n* = 3 in **c, d**). Error bars represent the SEM. Statistical significance was tested by two-sided one-way ANOVA with Tukey's test adjusted for multiple comparisons (**c, d**). Data are representative of three independent experiments (**c, d**).

S1263), Wnt ligands including Wnt1 (Abmart, EHM6128), Wnt2 (Abmart, EHM6133), Wnt3a (Peprotech, 315-20), Wnt5a (R&D, 645-WN), Wnt7b (Abmart, EHM6142), Wnt8b (R&D, 8419-WN), Wnt10b (R&D, 2110-WN), Norrin (R&D, 3497-NR-025), R-spondin (R&D, 7150-RS), PMA (MCE, HY-18739), Ionomycin (MCE, HY-13434), Brefeldin A (MCE, HY-16592), 5 μM 4-hydroxytamoxifen (MCE, HY-16950) and other drugs were added to cell culture medium in an optimal dose to treat ILC3s in vitro. Cell sorting was performed on the FACSAria (BD Bioscience).

### CHILP culture in vitro

OP9 stromal cells were cultured and irradiated in a cell culture plate in advance. CHILPs were isolated and purified from bone marrow of the donor mice, gated in 7AAD$^-$lineage$^-$CD127$^+\alpha_4\beta_7^+$CD25$^-$Flt3$^-$, stained with the following antibodies: 7AAD (BioLegend, 420404), CD127 (clone A7R34, BioLegend, 135023), $\alpha_4\beta_7$ (clone DATK32, BioLegend, 120608), CD25 (PC61, BioLegend, 102006), Flt3 (clone A2F10, BioLegend, 135312), CD45 (clone 30-F11, BioLegend, 103128) and lineage cocktail as mentioned before. CHILPs were cultured on OP9 cells with 20 ng/ml recombinant mouse IL-7 (Novoprotein, CC73) and 10 ng/ml recombinant mouse SCF (Novoprotein, C775) in Alpha-MEM medium including 15% FBS, 10 mM HEPES, 100 U/ml Penicillin, 100 μg/ml Streptomycin, 1 mM sodium pyruvate, 8 mg/ml Glutamine and 80 μM 2-Mercaptoethanol. 5 μM 4-hydroxytamoxifen (MCE, HY-16950) and other drugs were added to cell culture medium in an optimal dose to treat CHILPs in vitro. Cell sorting was performed on the FACSAria (BD Bioscience).

### Th17 and Treg cell culture in vitro

Naïve T cells were isolated by sorting CD4$^+$CD25$^-$CD62L$^{hi}$CD44$^{low}$ cells from the spleen and lymph nodes of wild-type mice, and activated with the plate-bound 7 μg/ml anti-CD3 (BioXCell, clone 145-2C11) and 7 μg/ml anti-CD28 (BioXCell, clone 37.51). Naïve T cells were stained with following antibodies: 7AAD (BioLegend, 420404), CD4 (clone GK1.5, BioLegend, 100451), CD25 (clone 3C7, BioLegend, 101912), CD62L (clone MEL-14, BioLegend, 104408), CD44 (clone IM7, BioLegend, 103012). Th17 cells were induced in RPMI 1640 medium including 10% FBS, 10 mM HEPES, 100 U/ml Penicillin, 100 μg/ml Streptomycin, 1 mM sodium pyruvate, 8 mg/ml Glutamine, 80 μM 2-Mercaptoethanol, and cytokine cocktails as indicated with the following reagents: IL-6 (Preprotech, 216-16, 20 ng/ml), TGF-β (R&D Systems, 240-B-010, 1 ng/ml), IL-1β (Peprotech, 211-11B, 10 ng/ml), IL-23 (R&D Systems, 1887-ML-010, 25 ng/ml). Treg cells were induced in RPMI 1640 medium including 10% FBS, 10 mM HEPES, 100 U/ml Penicillin, 100 μg/ml Streptomycin, 1 mM sodium pyruvate, 8 mg/ml Glutamine, 80 μM 2-Mercaptoethanol, 2 ng/ml TGF-β (R&D Systems, 7666-MB) and 10 ng/ml IL-2 (Peprotech, 212-12). Cell sorting was performed on the FACSARia (BD Bioscience).

### Flow cytometry analysis of ILC3s and T cells

Cells were stained for viability using the Fixable Viability Dye eFluor™ 780 (eBioscience™, 65-0865-14), followed by surface staining with varying combinations of the following antibodies in FACS buffer (1 × PBS + 2% FBS + 0.02% NaN₃): CD45 (clone 30-F11, BioLegend, 103128), CD3 (clone 17 A2, BioLegend, 100204), CD90.2 (clone 30-H12, BioLegend, 105335), CD127 (clone A7R34, BioLegend, 135023), CD4 (clone GK1.5, BioLegend, 100451), CD8a (clone 53-6.7, BioLegend, 100722), CCR6 (clone 29-2L17, BioLegend, 129814), Nkp46 (clone 29A1.4, BioLegend, 137612). For cytoplasmic staining, cells were fixed with IC Fixation Buffer (eBioscience™, 00-8222-49), and then cells were stained with IL-17A (clone TC11-18H10.1, BioLegend, 506904), IL-22 (clone 1H8PWSR, Thermo Fisher Scientific, 46-7221-82). For transcription factor staining, cells were fixed and permeabilized using Foxp3/Transcription Factor Fixation/Permeabilization Concentrate and Diluent (eBioscience™, 00-5523-00), and intracellular staining was performed for Foxp3 (clone 236 A/E7, Thermo Fisher Scientific, 17-

4777-42), RORγt (clone B2D, Thermo Fisher Scientific, 12-6988-82), GATA-3 (clone TWAJ, Thermo Fisher Scientific, 50-9966-42), T-bet (clone 4B10, Thermo Fisher Scientific, 644824), β-catenin (clone 15B8, Thermo Fisher Scientific, 12-2567-42), Ki-67 (SolA15, Thermo Fisher Scientific, 12-5698-82) and respective manufacturer isotype control antibodies. Flow cytometry was performed on Fortessa instruments (BD Biosciences) and analyzed with FlowJo software (V10).

### CFSE assay

A carboxyfluorescein diacetate succinimidyl ester (CFSE) stock (10 mM in DMSO, BD Biosciences) was thawed and diluted in 1 × PBS. For in vitro proliferation assay, sorted ILC3s isolated from Ctnnb1$^{ex3fl/wt}$ and CreER$^{T2}$Ctnnb1$^{ex3fl/wt}$ mice were incubated with 2 mM CFSE for 15 min at 37 °C. Then cells were washed twice with 1 × PBS and cultured in Alpha-MEM medium including 15% FBS, 10 mM HEPES, 100 U/ml Penicillin, 100 μg/ml Streptomycin, 1 mM sodium pyruvate, 8 mg/ml Glutamine and 80 μM 2-Mercaptoethanol. At different timepoints, the proliferation of ILC3s was examined based on the CFSE dilution rate by flow cytometry.

### Adoptive transfer of ILC3 and tamoxifen treatment

ILC3s were isolated and purified, then were transferred to recipient mice ($3 \times 10^5$ cells/mouse). Dissolve tamoxifen (Sigma-Aldrich) in corn oil (Sigma-Aldrich) at a concentration of 20 mg/ml by shaking overnight at 37 °C, then store it at 4 °C for the duration of injections. The recipient mice were intraperitoneally injected with tamoxifen (75 mg/kg body weight) once every day from day 1 to day 5.

### Histological assessment of intestine tissues

Colons were harvested from mice, flushed free of feces with ice-cold 1 × HBSS, formalin-fixed and Swiss rolled. All fixed organs were paraffin embedded and 4 μm sections were used for H&E staining. Embedding, sectioning, and H&E staining services were provided by Laboratory Animal Resources Center, University.

### Single-cell RNA-seq data analysis

Single-cell RNA sequencing (scRNA-seq) data of Ulcerative colitis was downloaded from the Single Cell Portal with accession number SCP259. Single-cell RNA sequencing data of colorectal cancer was downloaded from GEO with accession number GSE178341. The processed matrices of scRNA-seq data were imported using the scanpy package (version 1.9.0) in a separate analysis for each dataset[48]. The expression matrix was normalized by the total number of UMIs per cell and was log2-transformed. Cell types were annotated based on the previous studies[16,17], which served as a reference for assigning specific cell identities. Dot plot was used to investigate the activation of Wnt pathway-related genes in scRNA-seq datasets.

### RNA-Seq and data analysis

Living ILC3s were sorted by lineage$^-$CD90$^{high}$CD45$^{low}$ from the gut of adult Rag1$^{-/-}$ mice. Then cells were treated with DMSO or 20 μM CHIR-99021 for 24 hours in vitro. Next, cells were sorted again to exclude dead cells. The purified living ILC3s were collected and preserved in TRIzol (Invitrogen). All the RNA extraction, library preparation, and sequencing were done by The Beijing Genomics Institute (BGI). Differential gene testing was carried out by DESeq2. Comparisons between ILC3s treated by DMSO and CHIR-99021 were carried out. Gene Ontology (GO) analysis was done by the GO Enrichment analysis tool (http://geneontology.org/docs/go-enrichment-analysis/). The pot of PCA analysis, volcano plot, heatmap, and other plots were done by R.

### Western blot and co-immunoprecipitation

The protein level of GAPDH, TCF-1, β-catenin, JunB and NFATc2 in ILC3s was detected by Western blot with the following antibodies:

GAPDH (Beyotime, AF2819), TCF-1 (clone C63D9, Cell Signaling Technology, 2203), β-catenin (BD, 610154), JunB (clone C37F9, Thermo Fisher Scientific, PA1-835) and NFATc2 (clone 25A10.D6.D2, Thermo Fisher Scientific, MA1-025). In co-immunoprecipitation (co-IP) experiments, ILC3 pellets were resuspended in cell lysis buffer (Beyotime, P0013) including 1 mM PMSF (Beyotime, ST506). Then protein A + G magnetic beads (Beyotime, P2108), mouse IgG1 isotype control (clone G3A1, Cell Signaling Technology, 5415) and mouse anti-β-catenin (clone D10A8, Cell Signaling Technology, 8480) were used for IP. The IgG1-binding and β-catenin-binding proteins were further identified by Western blot.

### Luciferase assay

The region of P1–P4 around *Rorc* promoter was inserted into the firefly luciferase vector pGL3-vec. HEK293T cells (from Yang-Xin Fu, Tsinghua University, Beijing) were transfected with a control vector (pGL3-vec) or vector containing P1, P3, and P4, respectively (pGL3-P1, pGL3-P3, pGL3-P4), together with the renilla luciferase vector pRL-TK as an internal control. The cells were also cotransfected with empty vector (pCDNA-vec) or vector expressing FOXS1, KLF7, NFATc2, JunB, IRF7 (pCDNA-FOXS1, pCDNA-KLF7, pCDNA-NFATc2, pCDNA-JunB, pCDNA-IRF7) by Lipofectamine™ 3000 Transfection Reagent (Thermo Fisher Scientific). P1, P3, and P4 activity was measured by a dual-luciferase assay system (Promega) and results were normalized against the activity of the pRL-TK control group. The region of R1 (*Rorc* promoter), N1 (*Nfatc2* intron), and J1 (*Junb* promoter) was inserted into the firefly luciferase vector pGL3-vec. 293T cells were transfected with a control vector (pGL3-vec) or vector containing R1, N1, and J1, respectively (pGL3-R1, pGL3-N1, pGL3-J1), together with the renilla luciferase vector pRL-TK as an internal control. The cells were also cotransfected with empty vector (pCDNA-vec) or vector expressing TCF-1 (pCDNA-TCF-1) by Lipofectamine™ 3000 Transfection Reagent (Thermo Fisher Scientific). R1, N1, and J1 promoter activity was measured by a dual-luciferase assay system (Promega) and results were normalized against the activity of the pRL-TK control group.

### Transfection of ILC3s with siRNA

ILC3s were sorted from $Rag1^{-/-}$ mice and cultured in a 96-well round-bottom plate ($5 \times 10^4$ cells/well) in the presence of 20 ng/ml mIL-7. After 12 hours, 0.6 μg of siRNA that targeted *Nfatc2* (sense: 5′-CCA AUAAUGUCACCUCGAATT-3′, anti-sense: 5′-UUCGAGGUGACAUU AUUGGTT-3′) and *Junb* (sense: 5′-GCCUGUCUCUACACGACUATT-3′, anti-sense: 5′- UAGUCGUGUAGAGACAGGCTT-3′), respectively, were transfected into cells from each well with Lipofectamine™ 3000 transfection reagent (Thermo Fisher Scientific). Three days after transfection, the cells were harvested for analysis with flow cytometry or real-time RT-PCR.

### Quantitative real-time PCR

Total RNA of tissues and cells was purified using the AxyPrep Multisource Total RNA Miniprep Kit (Axygen) based on the protocol of the manufacturer, and cDNA was prepared using the RevertAid First Strand cDNA Synthesis Kit (Thermo Fisher Scientific). Real-time PCR was performed with the real-time PCR StepOnePlus system (Applied Biosystems), using Hieff™ qPCR SYBR Green Master Mix (YEASEN). Data were always normalized to *Gapdh* or *Actb* and analyzed by the $2^{-\Delta\Delta C}T$ method. The sequence of qPCR primers is shown in Supplementary Table 1.

### Assay for transposase accessible chromatin with high-throughput sequencing (ATAC-seq) assay

ATAC-seq assay was performed using the Novoprotein Chromatin Profile Kit for Illumina® according to the manufacturer's instructions. Briefly, around $2 \times 10^4$ purified living ILC3s were sorted and cells were centrifuged at $500 \times g$ at 4 °C for 5 min, washed with $1 \times$ PBS, and

centrifuged again. Cells were resuspended in lysis buffer and immediately centrifuged at $500 \times g$ at 4 °C for 5 min. The nuclei were at the bottom and the supernatant was removed. Pellets were resuspended in transposome mix and incubated at 37 °C for 30 min. Then, a stop buffer was added to the mixture at 55 °C for 5 min to stop the tagmentation. Next, DNA fragments were extracted and purified by the magnetic beads. At last, a DNA library was established and used for sequencing.

### Cleavage under targets and tagmentation (CUT&Tag) assay

The CUT&Tag assay was performed using YEASEN NovoNGS® CUT&Tag 3.0 High-Sensitivity Kit (for Illumina®) according to the manufacturer's instructions. Briefly, $1 \times 10^5$ ILC3s were washed twice with 1.5 mL of wash buffer and then mixed with activated concanavalin A beads. After successive incubations with the primary antibody (room temperature, 1 h) and secondary antibody (room temperature, 1 h), the cells were washed and incubated with pAG-Tn5 enzyme for 1 hour. Then, the MgCl₂ solution was added to activate tagmentation for 1 h. Then the tagmentation reaction was stopped, and the chromatin complex was digested with a solution containing 10 μL of 0.5 M EDTA, 3 μL of 10% SDS solution, and 2.5 μL of 20 mg/mL Proteinase K at 55 °C for 2 h. Next, DNA fragments were extracted and purified by the magnetic beads. At last, a DNA library was established and used for sequencing. The assays were accomplished with the following antibodies: Rabbit IgG Isotype Control (Abcam, ab37415), mouse IgG1 isotype control (clone G3A1, Cell Signaling Technology, 5415), Anti TCF-1 (clone C63D9, Cell Signaling Technology, 2203), Anti Phospho-Rpb1 CTD (Ser5) (clone D9N5I, Cell Signaling Technology, 13523), Anti Tri-Methyl-Histone H3 (Lys4) (clone C42D8, Cell Signaling Technology, 9751), Anti Acetyl-Histone H3 (Lys27) (clone D5E4, Cell Signaling Technology, 8173), Anti JunB (clone C37F9, Thermo Fisher Scientific, PA1-835), and Anti NFATc2 (clone 25A10.D6.D2, Thermo Fisher Scientific, MA1-025).

### ATAC-seq and CUT&Tag data analysis

Sequenced CUT&Tag and ATAC datasets were mapped to the mouse mm10 genome with Bowtie2 while CUT&Tag data were also additionally mapped to the *E. coli* genome. CUT&Tag reads were normalized by *E. coli* spike-in and were presented by Integrative Genomics Viewer software. Normalized reads were used as an input file for MACS2 callpeak with a cutoff $q < 0.05$ and IgG as a reference. Open chromatin (ATAC-seq) peaks were called with MACS2 using the '--nomodel' option and no background was provided followed by differential accessibility analysis with the csaw Bioconductor package. Motif enrichment was performed on peaks from CUT&Tag and ATAC and then annotated with annotatePeaks.pl in HOMER. The density plot and heatmap of peak distribution on the genome were calculated by deepTools (3.5.4) in Python.

### Immunofluorescence

The colons from experimental mice were fixed by 4% paraformaldehyde (PFA) fix solution (Beyotime), and then dehydrated by 30% sucrose overnight (Sigma). The colons were embedded in an optimal cutting temperature compound (SAKURA Tissue-Tek® O.C.T. Compound) and frozen at −80 °C. The embedded colons were cut into 10 μm-thick sections and blocked with $1 \times$ PBS + 0.2% Tween-20 (Sigma) + 5% FBS (Gibco) at room temperature for at least 2 hours, then incubated directly with indicated antibodies (1/200 dilution) at 4 °C overnight. Slides were washed in $1 \times$ PBS + 0.2% Tween-20 at room temperature with shaking for 10 min from 3 times. Slides were stained for 10 min with DAPI (Invitrogen, D1306), and then mounted with Fluoromount-G (SouthernBiotech). Antibodies against the following mouse antigens were used: EpCAM (clone 323/A3, Thermo Fisher Scientific, MA5-12436), E-cadherin (clone DECMA-1, Thermo Fisher Scientific, 53-3249-82), Ki-67 (clone SolA15, Thermo Fisher Scientific,

12-5698-82), CD45 (clone EP322Y, Abcam, ab40763). Slides were analyzed on an Olympus FV3000 microscope. The raw data were then analyzed by OlyVIA software (4.1.1).

### Quantification and statistical analysis

Two-way ANOVA with Sidak correction was used for weight change comparison. The log-rank Mantel−Cox test was employed to determine any statistical difference between the survival curves of the two groups. Where appropriate, other statistical analyses were performed using unpaired two-sided Student's t-test, unpaired two-sided Mann−Whitney U test, one-way ANOVA with Tukey's test, and two-way ANOVA with Sidak correction for multiple comparisons. All statistical tests were justified as appropriate and data met the assumptions of the test. All statistical tests were performed with GraphPad Prism 8.4.3 program. Data from such experiments are presented as mean values ± SEM; $p < 0.05$ was considered significant.

### Reporting summary

Further information on research design is available in the Nature Portfolio Reporting Summary linked to this article.

### Data availability

Data associated with all figures and tables is provided in the Source data file and Supplementary files. The raw data generated in this study have been deposited on NCBI under SRA accession: under accession code PRJNA976815. The processed CRC public scRNA-seq dataset used in this study is available in the Gene Expression Omnibus (GEO, https://www.ncbi.nlm.nih.gov/geo/) database under accession code GSE178341 and the processed Ulcerative colitis public scRNA-seq dataset used in this study are available in the Single Cell Portal with accession number SCP259. Source data are provided with this paper.

### Code availability

All code generated for analysis is available from the author upon request.

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

## Acknowledgements

We are grateful to Dan R. Littman (New York University, NY) for *Rorc*^cre mice; Chen Dong (Tsinghua University, Beijing) for CD45.1 mice; Xiaoyu Hu (Tsinghua University, Beijing) for *CreER*^T2 mice; Sheng Ding (Tsinghua University, Beijing) for OP9 stromal cells. We also thank the Core Facility of the Institute for Immunology and the Animal Facility at Tsinghua University for their support. This work was supported by the National Key R&D Program of China (2021YFC1712904, 2023YFC2306202, 2017YFA0103602), Beijing Natural Science Foundation (Z210015), the National Natural Science Foundation of China (82150104, 82122030, 32170872, 82141201, 31821003, and 91642106), and Innovation Team and Talents Cultivation Program of National Administration of Traditional Chinese Medicine (No. ZYYCXTD-D-202001). The Guo laboratory was also supported by the Research Fund, Vanke School of Public Health, Tsinghua University, and Center for Life Sciences, the Institute for Immunology, Tsinghua University.

## Author contributions

X.G. is the senior and corresponding author. X.G. conceived and designed the study. X.G. and J.H. prepared the manuscript. J.H. performed all experiments and assisted in data analysis. C.L. performed most of the sequencing data analysis. Z.G. participated in some experiments. X.Y. provided *Ctnnb1*^ex3flox/flox mice. X.L. provided helpful suggestions on bioinformatics analysis.

## Competing interests

The authors declare no competing interests.
