## [Peer Review File · Nature Communications]

REVIEWER COMMENTS

Reviewer #1 (Remarks to the Author):

In this study the authors report that activation of Wnt/b-catenin signalling in ILC3 results in loss of RORgt expression, cell death and dysregulation of ILC3 transcriptional signatures driven in part via epigenetic changes within the RORgt locus - associated with a transcription factor network that involves TCF-1 and JunB. The authors infer that activation of this pathway in the context of IBD or colorectal cancer could explain the loss of protective ILC3 pathways that drive disease onset and progression, although direct evidence for this hypothesis is limited. Overall the study provides novel insights into the transcriptional and epigenetic regulation of ILC3, although there are questions regarding the activity of this pathway in vivo and relevance in health and disease that are not fully addressed.

Specific points

- Significant emphasis is placed upon the potential link between loss of ILC3 populations and/or function in IBD and CRC and activation of the Wnt/b-catenin pathway. While this is an interesting hypothesis and may be likely based on prior associations, little in the way of direct evidence for this is provided. In particular, Figure 1A comprises the major supporting evidence which is restricted to transcription of *Cttnb1* and *Tcf7*. In the case of CRC these transcripts are present in both normal tissue and tumor so it's hard to support such strong interpretations and conclusions linking b-catenin pathway and loss of ILC3 in disease without further development. Either the authors should tone down their interpretation and conclusions or provide further evidence to show direct activation of b-catenin pathway does occur in situ in this context.
- Similarly, while the authors utilize a number of approaches (pharmacological agonism and gain of function mouse model) to elegantly demonstrate the consequences of chronically activating this pathway, a limitation of the study is the absence of a loss of function approach to provide evidence of Wnt/b-catenin engagement in health or disease in vivo. As the authors mention in the discussion, a loss of function would be predicted to prevent activation of this pathway in inflammation and disease and rescue ILC3 function, which would have significant translational impact. If such an approach is not possible, evidence as to whether this pathway is indeed engaged by ILC3 in vivo by endogenous ligands in certain contexts (e.g. inflammation model) would be provide more physiological relevance that mature tissue-resident cells do indeed receive such signals in the tissue.
- There are a number of data sets where the gating strategy used for sorting or analysis is not sufficiently rigorous to attribute to ILC3 only (Lineage neg, CD90hi CD45 low), particularly in absence of RORgt-gfp or surface markers of ILC subsets. At the very minimum the authors must provide data to demonstrate that such a gating strategy is sufficient to gate only RORgt+ ILC and exclude other ILC subsets (e.g. ILC2, ILC1), both in control animals but also in flox or treatment groups where other cells could feasibly be altered and lose CD45 intensity thus, contaminating this gate.
- More generally, there are potential issues associated with the methodology and analysis of ILC3 across the study. As noted by the authors the "ILC3" used for bulk functional, transcriptional and epigenetic studies are highly heterogeneous containing at least three transcriptionally and epigenetically distinct cell types (i.e. CCR6+ ILC3, NCR+ ILC3 and DN ILC3). This leads to issues with interpretation of multiple

data sets as it is often difficult to discount preferential enrichment or effects on one ILC3 subset or another as major drivers of transcriptional, epigenetic or proliferative/survival changes. This is similarly problematic when performing bulk genomic assessment of a highly heterogeneous population. While I have no reason to doubt the overall findings and the pathways they highlight this does present a significant methodological caveat and should be extensively discussed and highlighted as such in the manuscript.

- A major effect of engaging the b-catenin pathway in ILC3 ex vivo is cell death. Can the authors rule out toxicity of the agonist? Alternatively how can changes to transcriptional signatures be distinguished from cell death and preferential loss of one ILC3 subset over another?

- On similar lines on a number of occasions the authors measure cytokine production in “ILC3” from animals where these cells should be absent – for example Figures 3g vs 3i, S2i vs S2k. It is unclear how such an analysis was possible given the lack of ILC3 in the conditional flox mice in this system – presumably this is analysis of total ILC for cytokine, not ILC3 – however that is not what is indicated in the figure legends or text.

- Figure 4 –ILC3 subsets have significantly different proliferative capacities, thus further resolution is required when studying the cells that show b-catenin activation and Ki-67/CFSE dilution. Addition of NKp46 and CCR6 to these analyses would be useful in determining whether ILC3 subsets see these signals and/or respond similarly in this system in regards to signalling, proliferation and cell death.

- The effects of Wnt/b-catenin activation are attributed in part to TCF-1 regulation of JunB and RORgt. What are the levels of TCF-1 expression in resting ILC3 subsets and to what degree does b-catenin agonism modulate that?

- The use of the CreERT2 model is an elegant and useful approach to distinguish developmental defects from agonism of the Wnt/b-catenin pathway in mature cells. It would be important to show tamoxifen administration to these animals in vivo can in part recapitulate the phenotypes and changes in ILC3 demonstrated ex vivo as ILC are difficult to work with long term ex vivo while many in vivo signals are lacking. Thus approach would further support the function of this pathway in a more physiologically representative setting.

- In many data sets loss of RORgt+ cells is also associated with an increase in GATA-3 and RORgt negative cells (e.g. Fig 2a, 3g, S1a+c, S2i). While the authors focus their interpretation on the cell death and loss of ILC3 in clinical setting, an alternative interpretation is that Wnt/b-catenin engagement results in a loss of mature ILC programmes and the development of a TCF-1 hi stem-like/progenitor state. This should be more fully considered in the discussion, particularly in the absence of clear evidence that mature ILC engage this pathway under homeostatic or disease settings.

Minor points

- There were a number of small typographical and grammatical errors that should be corrected (e.g. “health donors” in abstract, instead of healthy). Similarly the word “destroy” in the abstract may not be appropriate. Overall the manuscript is well written but further proof reading may help improve during revision.

- RORgt staining in Fig1h-l amongst Treg does not look robust and appears to largely be a compensation artefact (compare with typical RORgt+ populations in primary Treg e.g. 2g). Would suggest this data is removed or replaced.

- Similarly, Fig1 suggests Wnt agonism increases RORgt amongst Treg in an intrinsic manner. The data provided in Fig 2g-h ostensibly shows the same in Cre-positive Ctnnb1ex3fl mice however this appears to be a function of how the data is analysed, as the RORgt+ fraction of FoxP3+ Treg is actually less in RORc Cre x Ctnnb1ex3fl mice than the control counterparts (e.g. 12% of 45% = 26% of Treg, 6% of 18% = 33% of Control Treg) – and the major effect appears to be an increase in total FoxP3+ cells (or loss of FoxP3- cells) when this pathway is engaged. This analysis (i.e. % of RORgt amongst FoxP3+ cells) should be included as well for accuracy and completion and the discrepancies discussed.

- How can the authors reconcile complete loss of ILC3 in the RORc Cre x Ctnnb1ex3fl mice with reconstitution of the ILC3 compartment by RORc Cre x Ctnnb1ex3fl bone marrow in a mixed bone marrow chimera setting (Figure 2j,k). While these cells are relatively outcompeted by wild type cells they do provide progeny – suggesting they are not unable to give rise to ILC3, despite the fact no mature ILC3 are found in the tissues of these animals.

Reviewer #2 (Remarks to the Author):

In this study, the authors investigated how Wnt/ β -catenin signaling regulates ILC3 and intestinal inflammation. Although the first part showing Wnt/ β -catenin negatively regulates ILCs is interesting, the later parts, which are trying to explain how Wnt/ β -catenin mechanistically inhibits ILC3 and linking Wnt/ β -catenin regulation of intestinal inflammation through ILC3, are not convincing and lack of data support. Furthermore, some major concerns need to be addressed to improve the quality of the manuscript:

1) Fig 1 should also show Wnt/ β -catenin on Th17 cells in addition to Treg;

2) Fig 3, The RorcCreCtnnb1ex3fl/wt mice are not ILC3 specific. Their Th17 cells are also effected. Thus it is immature to contribute their phenotypes solely on ILC3. An approach to exclude Th17 cells are required to make such conclusion. In addition, need to show total colonic tissue IL-17 and IL-22 levels but not only ILC3 IL-17/IL-22 as intestinal Th17 cells produce more IL-17 and IL-22 in this model. Same is true for Citrobacter model.

3) Fig 5b. Although the data clearly indicated TCF-1 could bind

233 to three regions at Rorc locus (R1, R2, R3) in ILC3s, it is not clear why the author concluded "RORyt expression in ILC3s is not directly regulated by TCF-1 binding to Rorc locus"? It will be interesting to show how Wnt/ β -catenin differentially regulates Rorc in T cells and ILC3.

4) Figs 6 and 7. Although the data showed Wnt/ β -catenin/TCF-1 regulates NFATc2 and JunB, whether NFATc2 and JunB regulate Rorc is not convincing as only an association study. A definite approach is required to make such conclusion.

Reviewer #3 (Remarks to the Author):

In this manuscript, authors examined the mechanism of Wnt/ β -catenin signaling-induced ROR γ t downregulation in the intestinal ILC3 cells, which was associated with significant decrease of ILC3 in the model system. First, they found that β -catenin and TCF7 mRNA levels are increased in ILC3 cells of IBD or CRC tissues using database. Then, they hypothesized that Wnt signaling activation in ILC3 causes increased inflammatory responses and tumorigenicity. To assess this, authors constructed constitutive or conditional β -catenin stabilized ILC3 cells, and showed that Wnt activation is associated with ROR γ t upregulation and suppression of proliferation. As a possible underlying mechanism, they found that ROR γ t expression was suppressed by epigenetic mechanism in the β -catenin-stabilized ILC3 cells. Moreover, they also found that expression of JunB, that induces RORC expression, was downregulated by β -catenin/TCF in the ILC3 cells. Based on these results, they concluded that activation of Wnt/ β -catenin in the intestinal ILC3 cells inhibit ROR γ t expression, which leads to increased inflammatory responses and generation of tumorigenic microenvironment.

Although the genetic evidence that β -catenin stabilization in ILC3 cells promotes colitis phenotype is interesting, there are several significant concerns in the experimental design and interpretation of the data. Because of the following criticisms, the data do not support the conclusion that Wnt/ β -catenin signaling activation causes intestinal inflammation through ILC3 cell regulation.

1. The mRNA levels of CTNNB1 and TCF7 are increased in human IBD and CRC tissue ILC3 cells (Fig. 1a). However, the data do not necessarily indicate Wnt/ β -catenin signaling activation. Wnt signaling is regulated by β -catenin stabilization rather than CTNNB1 expression. Moreover, it is possible that CTNNB1 is upregulated by negative feedback mechanism. Thus, the study lacks evidence of Wnt/ β -catenin activation in ILC3 of IBD and CRC tissues.
2. Authors used GSK3 inhibitor (CHIR-99021) as “Wnt agonist” in several experiments (Fig. 1 and 4). However, inhibition of GSK3 kinase activity may cause modification of variety of cellular signaling other than Wnt signaling activation. To examine the effect of Wnt signaling activation, ILC3 cells should be treated with Wnt ligand and R-spondin. Because genetic alterations are not found in ILC3 in IBD, ligand-dependent Wnt activation should be examined.
3. Interestingly, stabilization of β -catenin in ILC3 promoted DSS-induced colitis phenotypes in mouse model (Fig. 3). DSS treatment induces ulcer in the colon mucosa, and inflammatory responses are induced till mucosa are repaired. Thus, repair response also affects the severity of colitis in this model. Authors need to examine in more detail about the mechanism of increased colitis phenotype whether repair responses are impaired by Wnt activation in ILC3. In other words, it is important to examine the possible mechanism for increased inflammatory responses in the DSS-treated mouse colon.
4. Authors claimed that Wnt/ β -catenin activation in ILC3 leads to ROR γ t downregulation by epigenetic mechanism as well as downregulation of NFATc2 and JunB. In contrast, it is shown that Wnt/ β -catenin signaling activates transcription of RORC in other cells. How such opposite molecular mechanisms are regulated in different cell types. This should be discussed.
5. Stabilization of β -catenin was conditionally induced by using CreER mice. Which promoter is used in the model, and where did authors obtain the strain? No information about this was provided in the manuscript.

The changes in the manuscript have been highlighted.

Reviewer #1 (Remarks to the Author):

In this study the authors report that activation of Wnt/b-catenin signalling in ILC3 results in loss of ROR γ t expression, cell death and dysregulation of ILC3 transcriptional signatures driven in part via epigenetic changes within the ROR γ t locus - associated with a transcription factor network that involves TCF-1 and JunB. The authors infer that activation of this pathway in the context of IBD or colorectal cancer could explain the loss of protective ILC3 pathways that drive disease onset and progression, although direct evidence for this hypothesis is limited. Overall the study provides novel insights into the transcriptional and epigenetic regulation of ILC3, although there are questions regarding the activity of this pathway in vivo and relevance in health and disease that are not fully addressed.

We thank the reviewer for their supportive comments on our work.

Specific points

- Significant emphasis is placed upon the potential link between loss of ILC3 populations and/or function in IBD and CRC and activation of the Wnt/b-catenin pathway. While this is an interesting hypothesis and may be likely based on prior associations, little in the way of direct evidence for this is provided. In particular, Figure 1A compromises the major supporting evidence which is restricted to transcription of *Cttnb1* and *Tcf7*. In the case of CRC these transcripts are present in both normal tissue and tumor so it's hard to support such strong interpretations and conclusions linking b-catenin pathway and loss of ILC3 in disease without further development. Either the authors should tone down their interpretation and conclusions or provide further evidence to show direct activation of b-catenin pathway does occur in situ in this context.

Answer: We thank the reviewer for the suggestion. Besides analyzing *CTNNB1* and *TCF7* expression in ILC3s with published scRNA-seq results, we have tried to detect β -catenin expression at protein level by immunofluorescence (IF) staining. However, since there are very few ILC3s in human IBD and CRC patient samples and it is difficult to get optimal pathological sections, it is difficult for us to get direct evidence of Wnt/ β -catenin activation in ILC3 of IBD and CRC tissues at protein level.

Meanwhile, as shown in **Fig. 1a** in the manuscript, the expression of not only *CTNNB1* and *TCF7*, but also the downstream genes of Wnt pathway (*VEGFA*, *MYC*), was higher in ILC3s from IBD and CRC tissues, which suggests that Wnt pathway successfully activate the transcription of downstream genes. Besides, we found a number of downstream genes of Wnt pathway by TCF-1 CUT&Tag assay. As shown in Fig. R1, most of those genes was upregulated in ILC3s from inflamed tissue or tumor compared to control, which further verified the activation of Wnt pathway and

downstream genes.

Together, we find more evidences which further support the activation of Wnt pathway in ILC3s. And we have also tone down our interpretation and conclusions in the revision (**Line 80-82, Line 363-365 and Line 448-449**).

Fig. R1: Wnt/ β -catenin pathway is activated in ILC3s from IBD and CRC patients. Relative expression level of indicated genes in ILCs from healthy (H) and inflamed (I) tissues of IBD patients, ILC3s from normal tissues (N) and colorectal tumors (T) of CRC patients. The size of dots represents the fraction of cells in group, and the color of dots represents the mean expression of genes in group.

- Similarly, while the authors utilize a number of approaches (pharmacological agonism and gain of function mouse model) to elegantly demonstrate the consequences of chronically activating this pathway, a limitation of the study is the absence of a loss of function approach to provide evidence of Wnt/ β -catenin engagement in health or disease in vivo. As the authors mention in the discussion, a loss of function would be predicted to prevent activation of this pathway in inflammation and disease and rescue ILC3 function, which would have significant translational impact. If such an approach is not possible, evidence as to whether this pathway is indeed engaged by ILC3 in vivo by endogenous ligands in certain contexts (e.g. inflammation model) would provide more physiological relevance that mature tissue-resident cells do indeed receive such signals in the tissue.

Answer: We thank the reviewer for the suggestion. We also realized that the mouse model of a loss of function is important for our research and we bought *Cttnb1*^{flox/flox} mice from The Jackson Laboratory (JAX: 004152). Unfortunately, we found the expression of β -catenin was not successfully depleted in *Rorc*^{cre}*Cttnb1*^{fl/fl} mice, and later we found out there is a mutation at loxp site of *Cttnb1* locus as shown in Fig. R2. Thus, we could not continue our research with this mouse strain. We plan to make a new strain with the vendor.

Fig. R2: The mutated sequence of 5' loxP site in *Ctnnb1*^{lox/lox} mice from The Jackson Laboratory.

Based on our previous research, it is quite difficult to perform gene editing directly in ILC3s by CRISPR/Cas9 system. On the other hand, we tried to transfect bone marrow cells to knockout *Ctnnb1* expression. However, we found that *Ctnnb1* KO bone marrow cells could not proliferate and develop into ILCs. Therefore, we treated ILC3 with various Wnt ligands to examine the effect of Wnt signaling activation. As shown in **Supplementary Fig. 1n-q**, we treated ILC3 with a combination of Wnt ligands (Wnt) and PMA+Ionomycin (P+I). Wnt ligands significantly inhibited ROR γ t expression in activated but not resting ILC3s, suggesting that Wnt signaling may have an effect on ILC3s during inflammation.

- There are a number of data sets where the gating strategy used for sorting or analysis is not sufficiently rigorous to attribute to ILC3 only (Lineage neg, CD90^{hi} CD45^{low}), particularly in absence of ROR γ t-gfp or surface markers of ILC subsets. At the very minimum the authors must provide data to demonstrate that such a gating strategy is sufficient to gate only ROR γ t⁺ ILC and exclude other ILC subsets (e.g. ILC2, ILC1), both in control animals but also in flox or treatment groups where other cells could feasibly be altered and lose CD45 intensity thus, contaminating this gate.

Answer: We thank the reviewer for the suggestion. Previously we have published the gating strategy (Methods in Molecular Biology, 2016)¹. The gating strategy of Lineage⁻CD90^{hi}CD45^{low} is used frequently for gut ILC3's sorting and over 95% of cells in the gate are ROR γ t⁺ ILC3s. The proportion of ROR γ t⁺ cells in the gate is shown in figure below.

We also performed ROR γ t staining in this gating strategy as shown in Fig. R3. Over 95% of cells in the Lineage⁻CD90^{hi}CD45^{low} gate are ROR γ t⁺ ILC3s from both

Ctnnb1^{ex3fl/wt} and *CreER*^{T2}*Ctnnb1*^{ex3fl/wt} mice.

Fig. R3: Intestinal lineage⁻CD90^{high}CD45^{low} LPLs are RORγt⁺ ILC3s. LPLs were isolated from gut of *Ctnnb1*^{ex3fl/wt} and *CreER*^{T2}*Ctnnb1*^{ex3fl/wt} mice and gated in lineage⁻CD90^{high}CD45^{low}. Then lineage⁻CD90^{high}CD45^{low} cells were further analyzed with the expression of RORγt by flow cytometry.

- More generally, there are potential issues associated with the methodology and analysis of ILC3 across the study. As noted by the authors the “ILC3” used for bulk functional, transcriptional and epigenetic studies are highly heterogeneous containing at least three transcriptionally and epigenetically distinct cell types (i.e. CCR6⁺ ILC3, NCR⁺ ILC3 and DN ILC3). This leads to issues with interpretation of multiple data sets as it is often difficult to discount preferential enrichment or effects on one ILC3 subset or another as major drivers of transcriptional, epigenetic or proliferative/survival changes. This is similarly problematic when performing bulk genomic assessment of a highly heterogeneous population. While I have no reason to doubt the overall findings and the pathways they highlight this does present a significant methodological caveat and should be extensively discussed and highlighted as such in the manuscript.

Answer: We thank the reviewer for the suggestion. We further analyzed the effect of Wnt/β-catenin signaling on ILC3 subsets. As shown in **Figure 2c-d and Supplementary Fig. 2e-g**, we found a universal effect of Wnt/β-catenin pathway on all ILC3s, including CCR6⁺ ILC3, NCR⁺ ILC3 and DN ILC3. Although the proportion of CCR6⁺ ILC3 decreased and the proportion of NCR⁺ ILC3 increased in *Rorc*^{cre}*Ctnnb1*^{ex3fl/wt} mice, the numbers of all ILC3 subsets were dramatically reduced. Thus, we thought that the calculation of ILC3 subsets' proportion was not so accurate. To determine whether Wnt/β-catenin signaling has different effects on ILC3 subsets, CCR6⁺, Nkp46⁺ and CCR6/Nkp46 double negative (DN) ILC3s were also sorted by flow cytometry, then were treated with 4-Hydroxytamoxifen and evaluated with cell viability and proliferation capability. As shown in **Supplementary Fig. 5a-d**, after β-catenin activation, every subset of ILC3s showed an increase in cell death and a

decrease in proliferation, similar to the general ILC3 response.

- A major effect of engaging the b-catenin pathway in ILC3 ex vivo is cell death. Can the authors rule out toxicity of the agonist? Alternatively how can changes to transcriptional signatures be distinguished from cell death and preferential loss of one ILC3 subset over another?

Answer: We thank the reviewer for the question. In order to exclude the possibility of Wnt agonist toxicity, ILC2s were also isolated and treated with CHIR-99021 at the same dose as ILC3s. As shown in **Supplementary Fig. 1f-m**, despite Wnt agonist significantly increasing β -catenin protein levels, GATA-3 expression and cell viability in ILC2s were not affected, suggesting that increased activation of Wnt/ β -catenin has limited effect on ILC2s and also indicate that the effect of Wnt agonist CHIR-99021 on ILC3s probably is not due to toxicity.

- On similar lines on a number of occasions the authors measure cytokine production in “ILC3” from animals where these cells should be absent – for example Figures 3g vs 3i, S2i vs S2k. It is unclear how such an analysis was possible given the lack of ILC3 in the conditional flox mice in this system – presumably this is analysis of total ILC for cytokine, not ILC3 – however that is not what is indicated in the figure legends or text.

Answer: We thank the reviewer for the reminder. These cells were gated in ILCs, not ILC3s. We have interpreted the data more clearly in the figure legends (revised **Fig. 3i-j** and **Supplementary Fig. 4k-l**).

- Figure 4 –ILC3 subsets have significantly different proliferative capacities, thus further resolution is required when studying the cells that show b-catenin activation and Ki-67/CFSE dilution. Addition of Nkp46 and CCR6 to these analyses would be useful in determining whether ILC3 subsets see these signals and/or respond similarly in this system in regards to signalling, proliferation and cell death.

Answer: We thank the reviewer for the suggestion. To determine whether Wnt/ β -catenin signaling has different effects on ILC3 subsets, CCR6⁺, Nkp46⁺ and CCR6/Nkp46 double negative (DN) ILC3s were also sorted from *CreER^{T2}Cttnb1^{ex3fl/wt}* mice by flow cytometry, then were treated with 4-Hydroxytamoxifen and evaluated with cell viability and proliferation capability. As shown in **Supplementary Fig. 5a-d**, although β -catenin activation showed different levels of regulation on the three ILC3 subsets, all of them showed an increase in cell death and a decrease in proliferation, similar to the general ILC3 response.

- The effects of Wnt/b-catenin activation are attributed in part to TCF-1 regulation of JunB and ROR γ t. What are the levels of TCF-1 expression in resting ILC3 subsets and to what degree does b-catenin agonism modulate that?

Answer: We thank the reviewer for the question. The protein levels of TCF-1 expression in resting ILC3s and Wnt-activated ILC3 have been provided in **Fig. 5a** of our manuscript. Wnt agonist positively regulated β -catenin and TCF-1 expression in ILC3s.

- The use of the CreERT2 model is an elegant and useful approach to distinguish developmental defects from agonism of the Wnt/b-catenin pathway in mature cells. It would be important to show tamoxifen administration to these animals *in vivo* can in part recapitulate the phenotypes and changes in ILC3 demonstrated *ex vivo* as ILC are difficult to work with long term *ex vivo* while many *in vivo* signals are lacking. Thus approach would further support the function of this pathway in a more physiologically representative setting.

Answer: We thank the reviewer for the suggestion. To compare the potential different phenotypes and changes in ILC3 *in vivo* and *ex vivo*, the same number of $CreER^{T2}Cttnb1^{ex3fl/wt}$ ILC3s and control ILC3s were transferred into $Rag2^{-/-}Il2rg^{-/-}$ recipients and then treated with tamoxifen to activate Wnt pathway in ILC3s *in vivo*. As shown in **Supplementary Fig. 5e-f**, nearly half of $CreER^{T2}Cttnb1^{ex3fl/wt}$ ILC3s were positive for β -catenin⁺ at day 6. And there are fewer $CreER^{T2}Cttnb1^{ex3fl/wt}$ ILC3s than control ILC3s, which was consistent with the results obtained *in vitro*. Based on those findings, in a more physiologically representative condition, Wnt pathway induces the deficiency of ILC3s *in vivo*, and few β -catenin⁺ ILC3s could be found at the later stage.

- In many data sets loss of ROR γ t⁺ cells is also associated with an increase in GATA-3 and ROR γ t negative cells (e.g. Fig 2a, 3g, S1a+c, S2i). While the authors focus their interpretation on the cell death and loss of ILC3 in clinical setting, an alternative interpretation is that Wnt/b-catenin engagement results in a loss of mature ILC programmes and the development of a TCF-1 hi stem-like/progenitor state. This should be more fully considered in the discussion, particularly in the absence of clear evidence that mature ILC engage this pathway under homeostatic or disease settings.

Answer: We thank the reviewer for the suggestion. It has been reported TCF-1 plays an important role in ILC development². TCF-1 upregulation is essential for ILC progenitors in bone marrow². In mature ILC3, activation of the Wnt pathway results in ROR γ t downregulation and cell death. It is possible, however, that activated Wnt pathway upregulates TCF-1 expression and prevents stem-like ILC progenitors from undergoing differentiation into ILC3. Future studies should utilize TCF-1 reporter mice in order to examine those TCF-1^{high} stem-like progenitor cells, and then analyze their relationship with ILC3s. We added more discussion on **Line 418-422** in the Discussion section.

Minor points

- There were a number of small typographical and grammatical errors that should be corrected (e.g. “health donors” in abstract, instead of healthy). Similarly the word

“destroy” in the abstract may not be appropriate. Overall the manuscript is well written but further proof reading may help improve during revision.

Answer: We thank the reviewer for the suggestion. We have reviewed our entire manuscript and corrected the errors.

- RORgt staining in Fig1h-I amongst Treg does not look robust and appears to largely be a compensation artefact (compare with typical RORgt⁺ populations in primary Treg e.g. 2g). Would suggest this data is removed or replaced.

Answer: We thank the reviewer for the suggestion. We have included Th17 data in **Fig. 1g-k**, and moved the Treg data to **Supplementary Fig. 1a-e**. Actually, unlike the fresh isolated RORγt⁺Foxp3⁺ Treg cells from mouse intestine (**Fig. 2g**), the expression level of RORγt in iTreg cells induced from naïve CD4⁺ T cells *in vitro* is a little low, which is also reported by other groups (Quandt et al., Nature Immunology, 2021) as shown below³.

Ex vivo stabilization of β-catenin in human Treg cells induces the pro-inflammatory phenotype.

(a–d) *Ex vivo* treatment of HD PBMCs with GSK-3β inhibitor Chiron for 4 to 7 d. Flow cytometric histograms and cumulative MFI analysis of β-catenin (a and b) and RORγt (c and d) expression in CD25⁺Foxp3⁺ Treg cells in these cultures. DMSO, dimethylsulfoxide.

- Similarly, Fig1 suggests Wnt agonism increases RORgt amongst Treg in an intrinsic manner. The data provided in Fig 2g-h ostensibly shows the same in Cre-positive Ctnnb1^{ex3fl} mice however this appears to be a function of how the data is analysed, as the RORgt⁺ fraction of FoxP3⁺ Treg is actually less in RORc Cre x Ctnnb1^{ex3fl} mice than the control counterparts (e.g. 12% of 45% = 26% of Treg, 6% of 18% = 33% of Control Treg) – and the major effect appears to be an increase in total FoxP3⁺ cells (or loss of FoxP3⁻ cells) when this pathway is engaged. This analysis (i.e. % of RORgt amongst FoxP3⁺ cells) should be included as well for accuracy and completion and the discrepancies discussed.

Answer: We thank the reviewer for the suggestion. As shown in **Fig. 2g-h**, although the proportion of RORγt⁻FoxP3⁺ and RORγt⁺FoxP3⁺ CD4⁺ T cells was increased in *Rorc*^{cre}*Ctnnb1*^{ex3fl/wt} mice, the proportion of RORγt⁺FoxP3⁺ cells in total FoxP3⁺ Treg cells was indeed partly decreased in *Rorc*^{cre}*Ctnnb1*^{ex3fl/wt} mice, which is not consistent with the *in vitro* data (revised **Supplementary Fig. 1a-e**). Considering that the development of RORγt⁺FoxP3⁺ T cells *in vivo* is also influenced by other T cells and

ILC3s, which are also changed in *Rorc*^{cre}*Ctnnb1*^{ex3fl/wt} mice, our data suggest the Treg phenotype *in vivo* maybe involved both intrinsic and extrinsic mechanisms. The discussion has been added in the revision (**Line 130-135**).

- How can the authors reconcile complete loss of ILC3 in the RORc Cre x *Ctnnb1*^{ex3fl} mice with reconstitution of the ILC3 compartment by RORc Cre x *Ctnnb1*^{ex3fl} bone marrow in a mixed bone marrow chimera setting (Figure 2j,k). While these cells are relatively outcompeted by wild type cells they do provide progeny – suggesting they are not unable to give rise to ILC3, despite the fact no mature ILC3 are found in the tissues of these animals.

Answer: We thank the reviewer for the question. We think that in bone marrow chimera model, the recipient mice were all lethally irradiated and the majority of immune cells were generated from transferred bone marrow cells. Since *Rorc*^{cre}*Ctnnb1*^{ex3fl/wt} mice only delete the exon 3 of *Ctnnb1* in ROR γ ⁺ ILC3s but not ROR γ ⁻ ILC progenitors, ILC3s newly generated from *Rorc*^{cre}*Ctnnb1*^{ex3fl/wt} bone marrow could be found. However, in adult mice at steady state, almost all of gut ILC3s are tissue-resident and the maintenance of gut ILC3s is dependent on local replenishment instead of bone marrow differentiation⁴. In adult *Rorc*^{cre}*Ctnnb1*^{ex3fl/wt} mice, accumulated cell death and inhibited proliferation led to ILC3' deficiency. To confirm that, we isolated common helper ILC progenitors (CHILPs) from *Ctnnb1*^{ex3fl/wt} and *CreER*^{T2}*Ctnnb1*^{ex3fl/wt} bone marrow and detect ILC3 development *in vitro*. We found that both control and *CreER*^{T2}*Ctnnb1*^{ex3fl/wt} CHILPs could generate ILC3s, but the latter CHILPs generated fewer ILC3s than control group (**Supplementary Fig. 5g-i**), suggesting that β -catenin activation may also influence ILC3s development or maintenance *in vitro*.

Reviewer #2 (Remarks to the Author):

In this study, the authors investigated how Wnt/ β -catenin signaling regulates ILC3 and intestinal inflammation. Although the first part showing Wnt/ β -catenin negatively regulates ILCs is interesting, the later parts, which are trying to explain how Wnt/ β -catenin mechanistically inhibits ILC3 and linking Wnt/ β -catenin regulation of intestinal inflammation through ILC3, are not convincing and lack of data support. Furthermore, some major concerns need to be addressed to improve the quality of the manuscript:

We thank the reviewer for their supportive comments on our work.

1) Fig 1 should also show Wnt/ β -catenin on Th17 cells in addition to Treg;

Answer: We thank the reviewer for the suggestion. We isolated naïve CD4⁺ T cells from WT mice and induced Th17 cell differentiation *in vitro*. We treated Th17 cells with CHIR-99021 to examine the effect of Wnt signaling activation on ROR γ t expression in Th17 cells. As shown in revised **Fig. 1g-k**, CHIR-99021 also upregulated ROR γ t in Th17 cells.

2) Fig 3, The *Rorc^{cre}Ctnnb1^{ex3fl/wt}* mice are not ILC3 specific. Their Th17 cells are also effected. Thus it is immature to contributes their phenotypes solely on ILC3. An approach to exclude Th17 cells are required to make such conclusion. In addition, need to show total colonic tissue IL-17 and IL-22 levels but not only ILC3 IL-17/IL-22 as intestinal Th17 cells produce more IL-17 and IL-22 in this model. Same is true for *Citrobacter* model.

Answer: We thank the reviewer for the suggestion. To further exclude the potential role of Th17 cells, we crossed *Rag1^{-/-}* mice with *Rorc^{cre}Ctnnb1^{ex3fl/wt}* mice to generate *Rag1^{-/-}Rorc^{cre}Ctnnb1^{ex3fl/wt}* mice and detect their phenotypes in DSS-induced colitis model. As shown in **Fig. 3o-p**, upon oral administration of DSS, *Rag1^{-/-}Rorc^{cre}Ctnnb1^{ex3fl/wt}* mice rapidly lost body weight and died around day 7, compared to much less body weight loss and no deaths among their littermate control mice. More severe diarrhea, colitis and colon pathology were also observed in *Rag1^{-/-}Rorc^{cre}Ctnnb1^{ex3fl/wt}* mice compared to control mice (**Fig. 3q-r**). Similarly, few ILC3s could be detected (**Supplementary Fig. 3c-d**) and cytokine production from ILC3 was impaired (**Supplementary Fig. 3e-f**) in the colon of *Rag1^{-/-}Rorc^{cre}Ctnnb1^{ex3fl/wt}* mice. All those data indicate that without the effect of T cells, activated Wnt pathway also results in the deficiency of ILC3 and exacerbates DSS-induced colitis.

We also detected IL-17A and IL-22 levels in total colonic tissues. Both IL-17A and IL-22 production was impaired in colon of *Rorc^{cre}Ctnnb1^{ex3fl/wt}* mice in DSS-induced colitis model (**Supplementary Fig. 3a**) and *C. rodentium* infection model (**Supplementary Fig. 4m**).

- 3) Fig 5b. Although the data clearly indicated TCF-1 could bind to three regions at *Rorc* locus (R1, R2, R3) in ILC3s, it is not clear why the author concluded "ROR γ t expression in ILC3s is not directly regulated by TCF-1 binding to *Rorc* locus"? It will be interesting to show how Wnt/ β -catenin differentially regulates *Rorc* in T cells and ILC3.

Answer: We thank the reviewer for the suggestion. Among the three regions (R1, R2, R3) at *Rorc* locus, R1 shares overlapping sequence with *Rorc* promoter and R2, R3 are in the intron of *Rorc*. Our data of luciferase assay suggested that the binding of TCF-1 to R1 might promote *Rorc* expression, while the TCF-1 to R2 and R3 remains to be studied by other methods. Thus, we would like to change the conclusion "ROR γ t expression in ILC3s is not directly regulated by TCF-1 binding to *Rorc* locus" into "ROR γ t expression in ILC3s is not directly inhibited by TCF-1 binding to *Rorc* promoter (R1)" in the description of Fig. 5 (Line 278-279). And we also added more discussion about the different mechanisms of ROR γ t expression in T cells and ILC3 in the Discussion section (Line 425-444).

- 4) Figs 6 and 7. Although the data showed Wnt/ β -catenin/TCF-1 regulates NFATc2 and JunB, whether NFATc2 and JunB regulate *Rorc* is not convincing as only an association study. A definite approach is required to make such conclusion.

Answer: We thank the reviewer for the suggestion. Unfortunately, we did not have NFATc2 KO and JunB KO mouse strains for further research. In our study, we transfected ILC3s with siRNA to know whether NFATc2 and JunB regulate *Rorc* (Fig. 6g-h). Knocking-down *Nfatc2* and *Junb* with siRNA could result in decreased ROR γ t expression in ILC3s at both protein and mRNA levels, suggesting that NFATc2 and JunB could indeed regulate ROR γ t expression in ILC3s. And, based on our previous research, it is quite difficult to transfect ILC3 with overexpressing plasmids. In the future, we plan to overexpress NFATc2 and JunB in the bone marrow cells from *Rorc*^{cre}*Ctnnb1*^{ex3fl/wt} mice to examine whether these two factors could rescue the ILC3 deficiency post β -catenin activation. On the other hand, we also plan to make ILC3-specific *Nfatc2* and *Junb* KO mice to confirm our observation.

Reviewer #3 (Remarks to the Author):

In this manuscript, authors examined the mechanism of Wnt/ β -catenin signaling-induced ROR γ t downregulation in the intestinal ILC3 cells, which was associated with significant decrease of ILC3 in the model system. First, they found that β -catenin and TCF7 mRNA levels are increased in ILC3 cells of IBD or CRC tissues using database. Then, they hypothesized that Wnt signaling activation in ILC3 causes increased inflammatory responses and tumorigenicity. To assess this, authors constructed constitutive or conditional β -catenin stabilized ILC3 cells, and showed that Wnt activation is associated with ROR γ t upregulation and suppression of proliferation. As a possible underlying mechanism, they found that ROR γ t expression was suppressed by epigenetic mechanism in the β -catenin-stabilized ILC3 cells. Moreover, they also found that expression of JunB, that induces RORC expression, was downregulated by β -catenin/TCF in the ILC3 cells. Based on these results, they concluded that activation of Wnt/ β -catenin in the intestinal ILC3 cells inhibit ROR γ t expression, which leads to increased inflammatory responses and generation of tumorigenic microenvironment.

Although the genetic evidence that β -catenin stabilization in ILC3 cells promotes colitis phenotype is interesting, there are several significant concerns in the experimental design and interpretation of the data. Because of the following criticisms, the data do not support the conclusion that Wnt/ β -catenin signaling activation causes intestinal inflammation through ILC3 cell regulation.

We thank the reviewer for their supportive comments on our work.

1. The mRNA levels of CTNNB1 and TCF7 are increased in human IBD and CRC tissue ILC3 cells (Fig. 1a). However, the data do not necessarily indicate Wnt/ β -catenin signaling activation. Wnt signaling is regulated by β -catenin stabilization rather than CTNNB1 expression. Moreover, it is possible that CTNNB1 is upregulated by negative feedback mechanism. Thus, the study lacks evidence of Wnt/ β -catenin activation in ILC3 of IBD and CRC tissues.

Answer: We thank the reviewer for the suggestion. Besides analyzing *CTNNB1* and *TCF7* expression in ILC3s with published scRNA-seq results, we have tried to detect β -catenin expression at protein level by immunofluorescence (IF) staining. However, since there are very few ILC3s in human IBD and CRC patient samples and it is difficult of get optimal pathological sections, it is difficult for us to get direct evidence of Wnt/ β -catenin activation in ILC3 of IBD and CRC tissues at protein level.

Meanwhile, as shown in **Fig. 1a** in the manuscript, the expression of not only *CTNNB1* and *TCF7*, but also the downstream genes of Wnt pathway (*VEGFA*, *MYC*), was higher in ILC3s from IBD and CRC tissues, which suggests that Wnt pathway successfully activate the transcription of downstream genes. Besides, we found a number of downstream genes of Wnt pathway by TCF-1 CUT&Tag assay. As shown in Fig. R1 below, most of those genes was upregulated in ILC3s from inflamed tissue or

tumor compared to control, which further verified the activation of Wnt pathway and downstream genes.

Together, we find more evidences which further support the activation of Wnt pathway in ILC3s. And we have also tone down our interpretation and conclusions in the revision (**Line 80-82, Line 363-365 and Line 448-449**).

Fig. R1: Wnt/ β -catenin pathway is activated in ILC3s from IBD and CRC patients. Relative expression level of indicated genes in ILCs from healthy (H) and inflamed (I) tissues of IBD patients, ILC3s from normal tissues (N) and colorectal tumors (T) of CRC patients. The size of dots represents the fraction of cells in group, and the color of dots represents the mean expression of genes in group.

2. Authors used GSK3 inhibitor (CHIR-99021) as “Wnt agonist” in several experiments (Fig. 1 and 4). However, inhibition of GSK3 kinase activity may cause modification of variety of cellular signaling other than Wnt signaling activation. To examine the effect of Wnt signaling activation, ILC3 cells should be treated with Wnt ligand and R-spondin. Because genetic alterations are not found in ILC3 in IBD, ligand-dependent Wnt activation should be examined.

Answer: We thank the reviewer for the suggestion. We also realized that the mouse model of a loss of function is important for our research and we bought *Ctnnb1*^{fl^{ox}/fl^{ox}} mice from The Jackson Laboratory (JAX: 004152). Unfortunately, we found the expression of β -catenin was not successfully depleted in *Rorc*^{cre}*Ctnnb1*^{fl/fl} mice, and later we found out there is a mutation at loxp site of *Ctnnb1* locus as shown in Fig. R2 below. Thus, we could not continue our research with this mouse strain. We plan to make a new stain with the vendor.

Fig. R2: The mutated sequence of 5' loxp site in *Ctnnb1*^{lox/lox} mice from The Jackson Laboratory.

Based on our previous research, it is quite difficult to perform gene editing directly in ILC3s by CRISPR/Cas9 system. On the other hand, we tried to transfect bone marrow cells to knockout *Ctnnb1* expression. However, we found that *Ctnnb1* KO bone marrow cells could not proliferate and develop into ILCs. Therefore, we treated ILC3 with various Wnt ligands to examine the effect of Wnt signaling activation. As shown in **Supplementary Fig. 1n-q**, we treated ILC3 with a combination of Wnt ligands (Wnt) and PMA+Ionomycin (P+I). Wnt ligands significantly inhibited ROR γ t expression in activated but not resting ILC3s, suggesting that Wnt signaling may have an effect on ILC3s during inflammation. However, we do not observe a significant change of β -catenin expression. We hypothesized that maybe the difference is too mild to detect, or Wnt ligands and P+I treatment might promote the translocation of β -catenin from cytoplasm to nucleus, which we could not detect due to the limitation of methods. In addition, as shown in the figure below, Quandt et al. (Nature Immunology, 2021) also reported that the combination of Wnt3a and anti-CD3/CD28 antibodies could slightly influence ROR γ t expression in Treg cells without changing the β -catenin level.

3. Interestingly, stabilization of β -catenin in ILC3 promoted DSS-induced colitis phenotypes in mouse model (Fig. 3). DSS treatment induces ulcer in the colon mucosa, and inflammatory responses are induced till mucosa are repaired. Thus, repair response also affects the severity of colitis in this model. Authors need to examine in more detail about the mechanism of increased colitis phenotype whether repair responses are impaired by Wnt activation in ILC3. In other words, it is important to examine the possible mechanism for increased inflammatory responses in the DSS-treated mouse colon.

Answer: We thank the reviewer for the suggestion. We also detected the expression of genes associated with inflammatory response and tissue repair in DSS-induced colitis model by qPCR. As shown in **Supplementary Fig. 3a, b**, the expression of ILC3-related genes *Reg3b*, *Reg3g*, *Il17a*, *Il22* and epithelial regeneration-related marker⁵ genes *Epcam*, *Tacstd2*, *Ly6a*, *Ly6g*, *Anxa1*, *Anxa8* in colon tissue was also impaired in *Rorc*^{cre}*Ctnnb1*^{ex3fl/wt} mice post DSS treatment, which indicated the disorder of gut

immunity and the impairment in tissue repair.

4. Authors claimed that Wnt/ β -catenin activation in ILC3 leads to ROR γ t downregulation by epigenetic mechanism as well as downregulation of NFATc2 and JunB. In contrast, it is shown that Wnt/ β -catenin signaling activates transcription of RORC in other cells. How such opposite molecular mechanisms are regulated in different cell types. This should be discussed.

Answer: We thank the reviewer for the suggestion. In the manuscript, we have showed that Wnt/ β -catenin pathway differentially regulated ROR γ t expression in different cell types, especially between T cells and ILC3s. Besides different epigenetic regulation, NFATc2 and JunB were two key upstream transcription factors which regulate ROR γ t expression not only in T cells^{6,7}, but also in ILC3. Surprisingly, activated Wnt/ β -catenin pathway does not change NFATc2 and JunB expression in T cells (Fig. R4). Those data indicate that Wnt pathway differentially regulates ROR γ t expression because of different transcriptional regulatory networks. Moreover, previous studies reported that TCF-1 directly promoted *Rorc* expression in T cells³. However, our findings showed opposite result in ILC3s. We hypothesize that distinct cell microenvironment results in different function of TCF-1 in different cell types. In the future, we plan to focus on how TCF-1 regulates ROR γ t expression in ILC3s. More discussion has been added in in the Discussion part of manuscript (Line 425-444).

Fig. R4: Activated Wnt/ β -catenin pathway differentially regulates NFATc2 and JunB expression in T cells and ILC3.

Th17, Treg cells and ILC3 were treated with DMSO and CHIR-99021 *in vitro*. The mRNA level of *Nfatc2* (a) and *Junb* (b) in Th17, Treg cells and ILC3 were detected by qPCR. Each dot represents one individual mouse. Error bars represent the SEM. Statistical significance was tested by unpaired two-tailed Student's t tests, **p < 0.01, ns, not significant. Data are representative of three independent experiments.

5. Stabilization of β -catenin was conditionally induced by using CreER mice. Which promoter is used in the model, and where did authors obtain the strain? No information about this was provided in the manuscript.

Answer: We thank the reviewer for the question. We have added the information in the manuscript. In our research, *CreER*^{T2} mice were from The Jackson Laboratory (JAX:

008463). *CreER^{T2}* was inserted after *Rosa26* promoter.

- 1 Guo, X., Muite, K., Wroblewska, J. & Fu, Y. X. Purification and Adoptive Transfer of Group 3 Gut Innate Lymphoid Cells. *Methods Mol Biol* **1422**, 189–196, doi:10.1007/978-1-4939-3603-8_18 (2016).
- 2 Yang, Q. *et al.* TCF-1 upregulation identifies early innate lymphoid progenitors in the bone marrow. *Nat Immunol* **16**, 1044–1050, doi:10.1038/ni.3248 (2015).
- 3 Quandt, J. *et al.* Wnt-beta-catenin activation epigenetically reprograms Treg cells in inflammatory bowel disease and dysplastic progression. *Nat Immunol* **22**, 471–484, doi:10.1038/s41590-021-00889-2 (2021).
- 4 Gasteiger, G., Fan, X., Dikiy, S., Lee, S. Y. & Rudensky, A. Y. Tissue residency of innate lymphoid cells in lymphoid and nonlymphoid organs. *Science* **350**, 981–985, doi:10.1126/science.aac9593 (2015).
- 5 Wang, Y. *et al.* Long-Term Culture Captures Injury–Repair Cycles of Colonic Stem Cells. *Cell* **179**, 1144–1159. e1115, doi:10.1016/j.cell.2019.10.015 (2019).
- 6 Yahia-Cherbal, H. *et al.* NFAT primes the human RORC locus for RORgammat expression in CD4(+) T cells. *Nat Commun* **10**, 4698, doi:10.1038/s41467-019-12680-x (2019).
- 7 Hasan, Z. *et al.* JunB is essential for IL-23-dependent pathogenicity of Th17 cells. *Nat Commun* **8**, 15628, doi:10.1038/ncomms15628 (2017).

REVIEWER COMMENTS

Reviewer #1 (Remarks to the Author):

Overall the authors have addressed my major concerns and in my opinion the findings are sufficiently robust and comprehensive to be of significant interest to the field.

However, there are several areas that could benefit from further clarification or discussion either in the text or by inclusion of control data to provide context, as detailed below. Importantly, the authors should compare and contrast their findings with a recent paper reporting the role of TCF-1 deletion in ILC3 (Zheng et al Cell Reports 2023) and consider to include a “limitations of the study” section to highlight aspects which were not yet demonstrated – in particular the physiological contexts in which b-catenin is likely to be similarly engaged in vivo as this is not demonstrated in an endogenous manner but would be important for the clinical implications and inferred links to human disease.

To be clarified in the text / discussion:

- The authors should reconcile how their findings fit with a recent paper investigating the role of TCF-1 in ILC3 (Zheng et al Cell Reports 2023). The latter would suggest TCF-1 is constitutively expressed by ILC3 and antagonises ROR γ t activity to suppress IL-17 production, but does not completely shut down ROR γ t expression or prevent ILC3 development. The authors write here in their discussion that “TCF-1 upregulation could impair ILC3 development” but these other recent findings would suggest that is not the case. Rather, do the authors suggest Wnt/beta-catenin activation causes my translocation of TCF-1 into the nucleus and binding to the ROR γ c locus?

- Similarly, many of the findings of the current manuscript use systems that constitutively activate this pathway, however the physiological context in which this would occur and/or the endogenous levels of activation seen both in health and disease in vivo in ILC3 are unclear. Is there a physiological situation where high levels of beta-catenin activation would be seen and lead to a loss of ILC3? This is inferred from the human disease context but not shown directly demonstrated in mouse models. This point should be discussed or acknowledged as a limitation of the current study.

- Similarly, is it possible that chronic activation leads to a threshold of activation that causes ILC3 loss, but that transient low-level interactions occurs continually and could simply antagonise ROR γ t and its programme without such drastic effects on ILC3 survival or development? In general the authors should be clear whether they hypothesise b-catenin prevents ROR γ t expression, or antagonises it's activity and suppresses the magnitude of its expression and/or ability to promote it's program.

Additional changes or clarifications to data:

- As above, in many cases the authors infer that ROR γ t is shut down/turned off by beta-catenin but in fact it appears more likely that post-development, transient engagement of this pathway reduces ROR γ t expression, but does not completely shut it off (e.g. Figure 1c vs h show ILC3 CHIR still have higher ROR γ t than T cell treated with DMSO). Throughout an addition of a control population known not to express ROR γ t (e.g. ILC2) would help provide important contextual information i.e. is the ROR γ t expression merely reduced, or completely lost. While FMO are provided these are often not informative for nuanced conclusions such as this due to some level of background staining being present even on

“negative cells”. More generally wording should be tempered where necessary in ex vivo assays where cells are transiently treated to be clear when RORgt is antagonised/reduced vs lost as this changes the interpretation slightly.

- In Supplementary Figure 1n-q the authors suggest that “Wnt signalling inhibited RORgt expression in activated but not resting ILC3s”. This point needs to be clarified as would it not be the case that many of the assays performed show effects in naïve mice or resting cells sorted and cultured ex vivo? Where cells are stimulated in ex vivo culture this needs to be clear in the text, and this point clarified in relation to work done in otherwise naïve animals. Similarly, the text could be clearer as to where assays are performed in vivo or ex vivo as on a few occasions this was not immediately apparent.

Minor points:

- Abstract: authors state dysregulation of ILCs has been found in IBD and CRC “yet without clear mechanisms”. While I understand the point, it is an overstatement to suggest changes in ILC3 in these diseases have not been investigated elsewhere and other mechanisms described, so language should be tempered.

Reviewer #2 (Remarks to the Author):

All my previous concerns have been addressed appropriately

Reviewer #3 (Remarks to the Author):

Authors responded to most of my comments, and the manuscript has been improved. However, several concerns have still remained as follows.

1. CHIR99021 is a GSK3 inhibitor, which activates canonical Wnt signaling by inhibition of beta-catenin phosphorylation, and thus this compound is Wnt activator but not Wnt agonist.
2. Therefore, key experiments that used only CHIR99021 should be confirmed by treatment of ILC3 cells with Wnt ligand, and presented in the main Figure(s).
3. DSS treatment induces ulcerative colitis, and mucosal repair process affects severity of inflammatory responses. Thus, histology of ulcerative colitis for both genotypes should be shown to compare the ulcer/repair phenotype and inflammatory responses in Fig. 3e.
4. CreER mice should be described as Rosa26-CreER at least in method section.

The changes in the manuscript have been highlighted.

Reviewer #1 (Remarks to the Author):

Overall the authors have addressed my major concerns and in my opinion the findings are sufficiently robust and comprehensive to be of significant interest to the field.

We thank the reviewer for their supportive comments on our work.

However, there are several areas that could benefit from further clarification or discussion either in the text or by inclusion of control data to provide context, as detailed below. Importantly, the authors should compare and contrast their findings with a recent paper reporting the role of TCF-1 deletion in ILC3 (Zheng et al Cell Reports 2023) and consider to include a “limitations of the study” section to highlight aspects which were not yet demonstrated – in particular the physiological contexts in which b-catenin is likely to be similarly engaged in vivo as this is not demonstrated in an endogenous manner but would be important for the clinical implications and inferred links to human disease.

Answer: We thank the reviewer for the suggestion. We have added a “limitations of the study” section in our manuscript. Relevant clarification and discussion have been provided in the section (Line 463-476).

To be clarified in the text / discussion:

- The authors should reconcile how their findings fit with a recent paper investigating the role of TCF-1 in ILC3 (Zheng et al Cell Reports 2023). The latter would suggest TCF-1 is constitutively expressed by ILC3 and antagonises ROR γ t activity to suppress IL-17 production, but does not completely shut down ROR γ t expression or prevent ILC3 development. The authors write here in their discussion that “TCF-1 upregulation could impair ILC3 development” but these other recent findings would suggest that is not the case. Rather, do the authors suggest Wnt/beta-catenin activation causes my translocation of TCF-1 into the nucleus and binding to the ROR γ c locus?

Answer: We thank the reviewer for the suggestion. The recent paper (Zheng *et al.*, Cell Reports, 2023) reported that TCF-1 is essential for the development of non-LTi ILC subsets including NCR+ILC3s, possibly by regulating the ILC progenitor. While TCF-1 is not required for LTi development, they found that it is able to promote LT expression in LTi cells for Peyer’s patch formation, and suppress IL-17 expression by LTi cells from the small intestine. Even though TCF-1 is constitutively expressed by ILC3s, few experimental evidence suggests that TCF-1 antagonizes ROR γ t activity to suppress IL-17 production in ILC3s. And in their paper the majority of the studies were carried out by knocking out *Tcf7* in hematopoietic cells or ILC3s, but without activating

or overexpressing TCF-1. Here, we focused on the consequences of activated Wnt/ β -catenin pathway in ILC3s to mimic the disease situation. We found that activated Wnt/ β -catenin pathway significantly upregulated TCF-1 expression, which could down-regulate ROR γ t expression and impair ILC3 development. Actually, Figure 2E of their study (Zheng *et al.*, Cell Reports, 2023), as shown below, indicates that TCF-1 deficiency in LTi cells leads to an increased ROR γ t expression, which is in agreement with our finding that TCF-1 over-activation results in decreased ROR γ t expression in ILC3s. Thus, we think that TCF-1 has a dose-dependent and stage-dependent effect on ROR γ t expression and ILC3 development and function. We have included a discussion about this part in our manuscript (Line 435-443).

Additionally, in our study, Western blot and CUT&Tag data showed that Wnt/ β -catenin activation causes increased TCF-1 expression and binding to the *Rorc* locus (Figure 5A-B). However, it remains unclear how TCF-1 directs *Rorc* transcription in ILC3, which warrants further investigation.

Figure 2E. Violin plots of MHCII, chemokine receptor *Ccr6* and *Cxcr5*, co-inhibitory receptor *Lag3* and *Pcd1*, and *Rorc* expression in WT and *Tcf7fl/flVavCre* LTi cells. (Zheng *et al.*, Cell Reports, 2023)

- Similarly, many of the findings of the current manuscript use systems that constitutively activate this pathway, however the physiological context in which this would occur and/or the endogenous levels of activation seen both in health and disease *in vivo* in ILC3 are unclear. Is there a physiological situation where high levels of beta-catenin activation would be seen and lead to a loss of ILC3? This is inferred from the human disease context but not shown directly demonstrated in mouse models. This point should be discussed or acknowledged as a limitation of the current study.

Answer: We thank the reviewer for the suggestion. In our study, we uncovered the potential role of dysfunctional Wnt/ β -catenin pathway in ILC3s, which may contribute to the intestinal diseases. We agreed with the reviewer that there is no direct evidence showing that high levels of β -catenin in a physiological situation leads to a loss of ILC3. Although we have shown that Wnt ligands treatment could inhibit ROR γ t expression *in vitro*, the upstream signals that activate the Wnt pathway in ILC3s in health and disease *in vivo* remain unclear. Therefore, we have included a discussion about this part in our manuscript (Line 463-476). In the future, we intend to determine the upstream signaling molecules of Wnt pathway in ILC3 at physiological state *in vivo*, as well as to develop appropriate mouse models for related studies.

- Similarly, is it possible that chronic activation leads to a threshold of activation that

causes ILC3 loss, but that transient low-level interactions occurs continually and could simply antagonise ROR γ t and its programme without such drastic effects on ILC3 survival or development? In general the authors should be clear whether they hypothesise b-catenin prevents ROR γ t expression, or antagonises it's activity and suppresses the magnitude of its expression and/or ability to promote it's program.

Answer: We thank the reviewer for the suggestion. Based on the current studies, we suggest that strongly activated Wnt/ β -catenin pathway inhibits ROR γ t expression, yet we still could not rule out the possibility that chronic activated Wnt/ β -catenin pathway also antagonizes ROR γ t activity and inhibits its downstream genes. A short-term or low-level stimulation may not produce such drastic results, but might inhibit ROR γ t function. Therefore, it would also be meaningful to distinguish the possible effect of Wnt/ β -catenin pathway on ROR γ t expression from transcriptional activity of ROR γ t. We have clarified this point and included a discussion about it in our manuscript (Line 463-476).

Additional changes or clarifications to data:

- As above, in many cases the authors infer that ROR γ t is shut down/turned off by beta-catenin but in fact it appears more likely that post-development, transient engagement of this pathway reduces ROR γ t expression, but does not completely shut it off (e.g. Figure 1c vs h show ILC3 CHIR still have higher ROR γ t than T cell treated with DMSO). Throughout an addition of a control population known not to express ROR γ t (e.g. ILC2) would help provide important contextual information i.e. is the ROR γ t expression merely reduced, or completely lost. While FMO are provided these are often not informative for nuanced conclusions such as this due to some level of background staining being present even on "negative cells". More generally wording should be tempered where necessary in ex vivo assays where cells are transiently treated to be clear when ROR γ t is antagonised/reduced vs lost as this changes the interpretation slightly.

Answer: We thank the reviewer for the suggestion. We agreed with the reviewer. As shown in Figure below, ROR γ t expression was detected in ILC2, DMSO-treated ILC3 and CHIR-treated ILC3 by flow cytometry. Here, the MFI of ROR γ t in ILC2 was significantly lower than that in ILC3, similar to the MFI in FMO control and isotype control groups. Based on those data, we think that ROR γ t expression was inhibited, but not completely shut off, by activated Wnt pathway. In our manuscript, we have tempered our description to describe the effect of Wnt pathway on ILC3.

RORγt expression in ILC2, DMSO-treated ILC3 and CHIR-treated ILC3.

- In Supplementary Figure 1n-q the authors suggest that “Wnt signalling inhibited RORγt expression in activated but not resting ILC3s”. This point needs to be clarified as would it not be the case that many of the assays performed show effects in naïve mice or resting cells sorted and cultured *ex vivo*? Where cells are stimulated in *ex vivo* culture this needs to be clear in the text, and this point clarified in relation to work done in otherwise naïve animals. Similarly, the text could be clearer as to where assays are performed *in vivo* or *ex vivo* as on a few occasions this was not immediately apparent.

Answer: We thank the reviewer for the suggestion. In our manuscript, upon isolation from mice, ILC3s were treated with IL-2 and IL-7 for survival, and different stimuli were added to the medium for research. There are indeed significant differences in the microenvironment of ILC3s *ex vivo* and *in vivo*. Various cytokines, chemokines and cell-cell interactions regulate ILC3’s function *in vivo*, whereas only IL-2 and IL-7 maintain ILC3’s survival *ex vivo*. In Supplementary Figure 1n-q (Figure 1r-u in 2nd revised manuscript), ILC3s were further treated with a combination of Wnt ligands (Wnt) and PMA+Ionomycin (P+I) *in vitro*. We found that Wnt ligands significantly inhibited RORγt expression in activated (P+I treated) ILC3s but not resting (control) ILC3s. Since our analysis was based on the notion that PMA+Ionomycin stimulates ILC3s, we concluded “Wnt signaling inhibited RORγt expression in activated but not resting ILC3s”. We have clarified this point and changed our description in our manuscript (Line 102-106).

Minor points:

- Abstract: authors state dysregulation of ILCs has been found in IBD and CRC “yet without clear mechanisms”. While I understand the point, it is an overstatement to suggest changes in ILC3 in these diseases have not been investigated elsewhere and other mechanisms described, so language should be tempered.

Answer: We thank the reviewer for the suggestion. We have changed the description into “Dysregulation of ILC3s has been found in the gut of patients with inflammatory bowel disease and colorectal cancer, yet the specific mechanisms still require more investigation.”

Reviewer #2 (Remarks to the Author):

All my previous concerns have been addressed appropriately

We thank the reviewer for their supportive comments on our work.

Reviewer #3 (Remarks to the Author):

Authors responded to most of my comments, and the manuscript has been improved. However, several concerns have still remained as follows.

We thank the reviewer for their supportive comments on our work.

1. CHIR99021 is a GSK3 inhibitor, which activates canonical Wnt signaling by inhibition of beta-catenin phosphorylation, and thus this compound is Wnt activator but not Wnt agonist.

Answer: We thank the reviewer for the suggestion. We have changed the name “Wnt agonist” to “Wnt activator” in our manuscript.

2. Therefore, key experiments that used only CHIR99021 should be confirmed by treatment of ILC3 cells with Wnt ligand, and presented in the main Figure(s).

Answer: We thank the reviewer for the suggestion. We agreed with the reviewer that the experiments should be confirmed by treating ILC3s with Wnt ligands. However, there are many kinds of Wnt ligands and receptors in human and mice, and there is potential for combinatorial diversity in the Wnt/ β -catenin signaling pathway, making it difficult to identify the physiologic role of specific receptor–ligand pairs. Additionally, as Wnt proteins are highly hydrophobic, only a few of them are currently commercially available. Therefore, it is quite difficult for us to determine which physiological Wnt ligand could activate β -catenin signaling in ILC3s. In Figure 1r-u of our manuscript, we found that a combination treatment of Wnt ligands and PMA+Ionomycin could significantly inhibits ROR γ t expression, suggesting that Wnt signaling may affect ILC3s in physiological situation. Further experiments are needed to determine the specific Wnt ligand that activates β -catenin signaling in ILC3s in the future. We have added a “limitations of the study” section to discussion this point in our manuscript (Line 463-476).

3. DSS treatment induces ulcerative colitis, and mucosal repair process affects severity of inflammatory responses. Thus, histology of ulcerative colitis for both genotypes should be shown to compare the ulcer/repair phenotype and inflammatory responses in Fig. 3e.

Answer: We thank the reviewer for the suggestion. To further evaluate the severity of DSS-induced colitis, we repeated the *in vivo* experiments and detected the ulcer/repair phenotype and inflammatory responses by immunofluorescence (IF) staining and HE staining. As shown in Figure 3e and Supplementary Figure 3a-b, the epithelium injuries are more severe in colons from *Rorc^{cre}Ctnnb1^{ex3fl/wt}* mice compared to control mice. It appears that there are fewer proliferative epithelial cells in *Rorc^{cre}Ctnnb1^{ex3fl/wt}* mice as indicated by Ki-67 staining, indicating impaired intestinal epithelial repair. A greater number of CD45⁺ cells are accumulated in colons of *Rorc^{cre}Ctnnb1^{ex3fl/wt}* mice, suggesting more immune cell infiltration and more severe inflammatory response. Overall, we analyzed the histology of ulcerative colitis for both genotypes and we find much more severe ulcer and inflammatory responses in *Rorc^{cre}Ctnnb1^{ex3fl/wt}* mice. We have clarified this point in our manuscript (Line 160-164).

4. CreER mice should be described as Rosa26-CreER at least in method section.

Answer: We thank the reviewer for the suggestion. We have added the related information of in method section.

REVIEWERS' COMMENTS

Reviewer #1 (Remarks to the Author):

The authors have addressed all my comments. The addition of new text in the discussion to discuss limitations and potential mechanistic insights will be useful for the readership.

Reviewer #3 (Remarks to the Author):

Authors adequately responded to all of my comments, and the manuscript has been significantly improved.

Reviewer #1 (Remarks to the Author):

The authors have addressed all my comments. The addition of new text in the discussion to discuss limitations and potential mechanistic insights will be useful for the readership.

We appreciate the reviewer for their helpful suggestions and supportive comments on our work.

Reviewer #3 (Remarks to the Author):

Authors adequately responded to all of my comments, and the manuscript has been significantly improved.

We appreciate the reviewer for their helpful suggestions and supportive comments on our work.